# Topology Optimization and Efficiency Evaluation of Short-Fiber-Reinforced Composite Structures Considering Anisotropy

Evgenii Kurkin *, Oscar Ulises Espinosa Barcenas, Evgenii Kishov and Oleg Lukyanov

Department of Aircraft Construction and Design, Samara National Research University, 34 Moskovskoe Shosse, Samara 443086, Russia; oscar.espinosa.barcenas@hotmail.com (O.U.E.B.); evgeniy.kishov@ssau.ru (E.K.); lukyanov.oe@ssau.ru (O.L.)
* Correspondence: kurkin.ei@ssau.ru; Tel.: +7-960-831-9009

**Abstract:** The current study aims to develop a methodology for obtaining topology-optimal structures made of short fiber-reinforced polymers. Each iteration of topology optimization involves two consecutive steps: the first is a simulation of the injection molding process for obtaining the fiber orientation tensor, and the second is a structural analysis with anisotropic material properties. Accounting for the molding process during the internal iterations of topology optimization makes it possible to enhance the weight efficiency of structures—a crucial aspect, especially in aerospace. Anisotropy is considered through the fiber orientation tensor, which is modeled by solving the plastic molding equations for non-Newtonian fluids and then introduced as a variable in the stiffness matrix during the structural analysis. Structural analysis using a linear anisotropic material model was employed within the topology optimization. For verification, a non-linear elasto-plastic material model was used based on an exponential-and-linear hardening law. The evaluation of weight efficiency in structures composed of short-reinforced composite materials using a dimensionless criterion is addressed. Experimental verification was performed to confirm the validity of the developed methodology. The evidence illustrates that considering anisotropy leads to stiffer structures, and structural elements should be oriented in the direction of maximal stiffness. The load-carrying factor is expressed in terms of failure criteria. The presented multidisciplinary methodology can be used to improve the quality of the design of structures made of short fiber-reinforced composites (SFRC), where high stiffness, high strength, and minimum mass are the primary required structural characteristics.

**Keywords:** multidisciplinary analysis and optimization; topology optimization; anisotropy; short fiber reinforced composites

## 1. Introduction

The aerospace and automotive industries have inherent requirements: high stiffness, high strength, and minimum mass are the primary concerns when designing structures [1–4]. For instance, high stiffness and minimal mass are required to limit the natural frequency of spacecraft to avoid resonance between the launch vehicle and itself [4–6], and in automobiles to improve performance while satisfying safety requirements [7,8]. Minimal mass is required for keeping the spacecraft structure lightweight to get to the desired orbital altitude by increasing the useful load fraction [9], lowering aircraft engineering costs by reducing airframe weight [10,11], and reducing aircraft and automobile operational costs by increasing fuel efficiency [10–12]. High strength is linked to the minimal mass requirement since materials with higher specific strength allow for keeping mass to a minimum. The same applies to the high stiffness requirement when using materials with high specific stiffness.

The strength of the structure depends on the structure's shape and material. Regarding materials, aluminum and titanium [13] are commonly used in the aerospace and automobile

industries due to their high specific strength and stiffness. Moreover, new materials have been researched for automobile [14] and aerospace applications [15–18], including ceramics [19], composites [20], and nanocomposites [21]. Furthermore, nanocomposites [22,23], shape memory polymers [24,25], and short fiber-reinforced composites (SFRC) have experienced an increase in aerospace applications in recent years [19,26–32]. The principal metrics for comparing materials, rather than Young's modulus or the ultimate tensile strength, are the specific stiffness and strength, as both mechanical characteristics are normalized by material density. Current strategies for designing efficient structures can be categorized into guidelines and recommendations [33], methods and methodologies [34–36], and even parametric [37–39] and topology optimizations [38,40–45]. Topology optimization (TO) has proven to be an excellent mathematical method for designing efficient structures and optimizing material arrangement within a given design space for a set of loads, boundary conditions, and constraints with the goal of minimizing a specific objective function. This method, also known as the variable density model, was originally proposed in the work [46], where it was applied to the optimal design of two-dimensional structures to overcome the computational difficulties associated with stress analysis of variable-thickness plates. Three-dimensional (3D) TO under strength and stiffness constraints, using material density as a design variable, was later presented in the studies [47–49]. Another approach to representing variable density using materials at the microstructure level and its application to TO problems can be found in the study [50], further explained in the work [51]. Designed structures are usually evaluated by considering maximum displacement, maximum strength, failure criteria values [52–54], or minimum mass of the structure, which are not reliable indicators since these values are local, except for mass, which is evaluated as an integral characteristic and does not provide comprehensive information about the structure other than its value. However, the load-carrying factor LCF G, proposed by Andrey Komarov in 1965 [46], allows us to relate both structural efficiency and weight requirements, facilitating objective comparisons between structures with relatively low levels of computational and modeling effort [55]. The fundamental physical meaning of the load-carrying factor is the integral characteristic of the structure, reflecting the internal forces in its elements (the mode of action of an external force on a structure up to its supports) and the extent of these internal forces (the extent of the external force transmission paths). A disadvantage of the LCF is its dimensional dependence; if the geometrical dimensions of the structure and external loads are changed, the magnitude of the LCF also changes, making it difficult to compare different structural arrangements. Therefore, the usage of the LCF coefficient, as proposed by Valery Komarov in the work [56], is more appropriate. In the accompanying article [57], a proposal is made to employ the method of mixtures for assessing the weight efficiency of composite structures, necessitating a detailed examination of composite materials at the microstructure level. From a practical standpoint, the utilization of strength criteria at the representative volume mesolevel appears to be more advantageous when evaluating complex-shaped structures for weight efficiency. In the present work, the strength criterion for homogenized composite material will be considered.

In summary, current modern materials such as SFRC, TO, metrics for assessing the structure shape as failure criteria, and LCF have contributed to increasing stiffness and strength while reducing the mass of structures. However, the interaction between them is often not carefully considered. The fiber orientation tensor is obtained by means of solving Folgar–Tucker's continuity equation, which is developed based on the fiber orientation kinetic theory of fiber suspensions [58–61]. SFRC is modeled as a transversely isotropic material, even though its stiffness depends not only on the material's mechanical characteristics but also on the fiber orientation tensor [62]; TO is typically performed in an isotropic medium [63,64], and LCF accounts for the equivalent stress of structures made of isotropic materials. In other words, anisotropy is either partially or completely ignored at various stages of the design process.

This work aims to develop a methodology for obtaining topology-optimal structures made of SFRC. The objectives of this work are as follows: to make the stiffness matrix de-

pendent on the fiber orientation tensor, to obtain the fiber orientation tensor by performing injection molding simulations, to formulate a metric for assessing SFRC structures that simultaneously evaluates the shape and material of the structure, and to experimentally verify the developed methodology. It is assumed that by considering the anisotropy during TO, the resulting topology will perform better than a topology obtained in an isotropic medium. Moreover, the anisotropy is directly related to the arrangement of fibers within the structure, specifically to the fiber orientation tensor, which is obtained by solving Folgar–Tucker's continuity equation [65] through modeling the injection molding process within the designated design region in Autodesk Moldflow. The consideration of anisotropy induced by the fiber arrangement is achieved by implementing the Advani-Tucker orientation averaging procedure on the material stiffness matrix [66] and attaching the resultant tensor to each mesh element. This procedure was realized with AnisoTopo [67]. The attachment of the stiffness matrix was accomplished by interpolating the meshes of the injection and structural models with Digimat MAP. Finally, TO was performed without altering the stiffness arrangement within the design region. The resultant topology can be assessed using a metric such as a modified LCF, which is described in terms of failure criteria for composite materials, such as the Tsai–Hill failure criterion.

The numerical and experimental results of this study confirm that considering the anisotropy during TO increases the stiffness of the resultant topologies. This is achieved by obtaining the fiber orientation tensor by solving the plastic molding equations for non-Newtonian fluids and making the stiffness matrix dependent on the obtained fiber orientation tensor.

## 2. Materials and Methods

### 2.1. Materials and Material Models

Three different materials and their non-linear models were considered (see Figure 1 and Table 1). These materials include two anisotropic materials: a 50% glass fiber reinforced polyamide 6, denoted as PA6 50GF [68], along with its associated model as presented in [69], and a 30% carbon fiber reinforced polyamide 6, referred to as PA6 30CF [70], with its corresponding model [71]. Additionally, an isotropic material, the D16T aluminum alloy [72], shares similarities with the 2024-T4 aluminum alloy [73]. It is worth noting that no specific preparation steps or treatments were applied to these materials before testing. The mechanical characteristics of these materials were determined through tensile tests conducted on samples cut from injected-molded plates with dimensions of 200 × 150 × 4 mm, following the ISO 527-2:2012(en) standard [74]. While the primary mechanical characteristics relevant to this study are summarized in Table 1, more comprehensive details on the material characteristics can be found in their respective documentation.

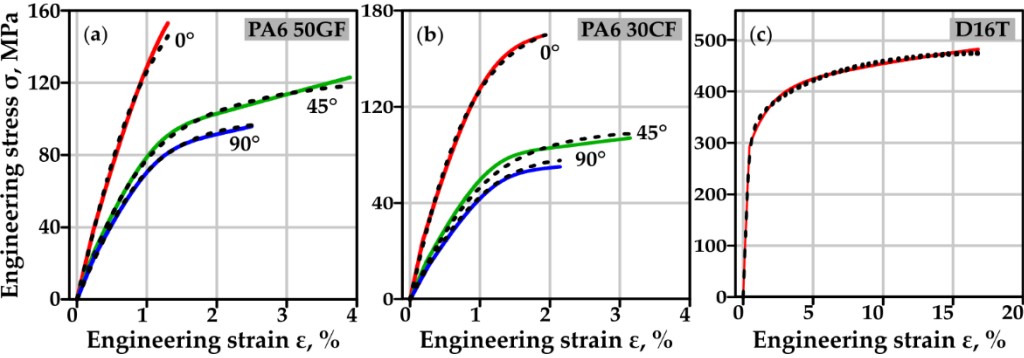

**Figure 1.** Stress–strain curve: (**a**) PA6 50GF; (**b**) PA6 30CF; and (**c**) D16T. Solid lines—model, dashed line—experiment. Angle between polymer flow and tension load: 0°—red, 45°—green, and 90°—blue.

**Table 1.** Material characteristics and mechanical properties.

| Characteristics | Material | | |
|---|---|---|---|
| | **PA 50GF** | **PA 30CF** | **D16T** |
| Matrix phase | | | |
| Matrix density, kg/m$^3$ | 1148 | | 2770 |
| Young's modulus, $E_m$ (MPa) | 4911 | 3994 | 66,059 |
| Poisson's coefficient, $\upsilon_m$ | 0.372 | 0.372 | 0.330 |
| Yield stress, $\sigma_y$ (MPa) | 17.21 | 14.5 | 294.48 |
| Hardening modulus, $R_\infty$ (MPa) | 37.1 | 37.00 | 109.51 |
| Hardening exponent, m | 371.21 | 458.30 | 75.72 |
| Linear hardening modulus, k (MPa) | 313.02 | 188.40 | 1107.60 |
| Reinforcement phase | | | |
| Fiber density, kg/m$^3$ | 2550 | 1780 | - |
| Young's modulus, $E_f$ (MPa) | 72,000 | 230,000 | - |
| Poisson's coefficient, $\upsilon_f$ | 0.22 | 0.20 | - |
| Fibers', AR | 13.58 | 16.54 | - |
| Wt. % | 30 | 50 | - |
| Material's ultimate tensile strength | | | |
| Longitudinal, X (MPa) | 153.31 | 169.35 | 476 |
| Transverse, Y (MPa) | 97.82 | 85.07 | - |
| Transverse shear strength, S (MPa) | 83.90 | 66.33 | - |

The material models for PA6 50GF and PA6 30GF were calibrated with experimental data as described in [71]. In summary, this calibration method involves modeling the material microstructure as a two-phase material and taking into account the fiber orientation in the calculation of mechanical properties. The material's mechanical properties were obtained by homogenizing the fibers and matrix using a second-order Mori–Tanaka homogenization scheme in Digimat MF. As the mechanical properties correspond to a unidirectional fiber-reinforced composite, it is modeled as a transversely isotropic material. Accounting for the fiber orientation was achieved by applying Tucker's averaging procedure. The matrix stress–strain state is described using the J2 plasticity model based on von Mises' equivalent stress. When the equivalent stress $\sigma_{eq}$ exceeds the yield stress $\sigma_y$, the stress–strain response becomes nonlinear, and plastic deformation occurs. The plastic's strength is then defined as follows:

$$\sigma_{plastic} = \sigma_y + R(\varepsilon_p), \tag{1}$$

where $\sigma_y$ is the yield stress; $R(\varepsilon_p) = k\varepsilon_p + R_\infty[1 - e^{-m\varepsilon_p}]$ represents the isotropic strain exponential and linear hardening law; and $\varepsilon_p$ is the accumulated plastic strain. Here, k is the linear hardening modulus in MPa; m is the hardening exponent; and $R_\infty$ is the hardening modulus in MPa. The material model parameters were adjusted by minimizing the difference between the tensile strain-stress curves of the composite material and the experimental results.

*2.2. Methods*

2.2.1. Topology Optimization

Topology optimization (TO) represents an effective approach for optimizing material distribution within a structure to enhance the transfer of internal forces from areas of load-carrying areas to support regions. Since the strain energy quantifies the energy stored in a body due to deformation, minimizing this response leads to a reduction in the body's compliance [49]. Strain energy is mathematically defined as W = 0.5 Fu, where F = Ku is the force expressed as the product of stiffness K and deformation u. The stiffness is influenced

by anisotropy resulting from the arrangement of fibers within the body, specifically the fiber orientation tensor.

In this study, the objective function of TO was to minimize the structure compliance by reducing the total strain energy of an anisotropic composite material. This was achieved by adjusting the topology density $\rho$ within the design region $\Omega$ while adhering to the given design region volume constraint $g_1(\mathbf{x})$ and constraints such as the "minimum member size" $g_2(\mathbf{x})$ and the "pull-out direction" $g_3(\mathbf{x})$. The formulation was defined as follows:

| minimize | $f(\mathbf{x}) = 0.5\mathbf{u}^T\mathbf{K}(\rho(\mathbf{x}), \mathbf{A}(\mathbf{x}))\mathbf{u},$ |
|---|---|
| by varying | $\rho(\mathbf{x}) \in \mathbf{(0, 1]}, \mathbf{x} \in \Omega,$ |
| subject to | $h(\mathbf{x}) = \mathbf{K}(\rho(\mathbf{x}), \mathbf{A}(\mathbf{x}))\mathbf{u} - \mathbf{F} = 0,$ |
| | $g_1(\mathbf{x}) = \int \rho(\mathbf{x})d\Omega - V_{ret} \leq 0,$ |
| | $g_2(\mathbf{x}) = \int |\nabla\rho(\mathbf{x})|d\Omega - \delta \leq 0,$ |
| | $g_3(\mathbf{x}) = \rho_i - \rho_k \leq 0 \qquad \forall\, x_i = x_k,\, y_i = y_k,\, |z_i| \geq |z_k|,$ |

Here, $\mathbf{K}$ represents the global stiffness matrix, $\mathbf{u}$ denotes the nodal displacement vector, F is the nodal force vector, and $\mathbf{x}$ is the vector containing design domain elements with coordinates x, y, and z. "Minimum member size" constraint $g_2(\mathbf{x})$ limits the change of the gradient of the density field with respect to spatial coordinates over the design domain. Thus, the minimum width of structural members on average becomes limited to a specified value. In addition, it is known [51] that the minimum member size constraint acts as a mesh independence filter for topology optimization. "Pull-out direction" constraint $g_3(\mathbf{x})$ provides a monotone decreasing of density $\rho(\mathbf{x})$ when moving away from a parting plane. $\mathbf{A}$ represents the fiber orientation tensor, and $\delta$ is related to the minimum structural member size. TO, in this work, was performed using the "Sequential Convex Programming" solver within the Ansys Mechanical Workbench 18.2 software. The relationship between topological density and finite element stiffness matrix is carried out by the SIMP (Solid Isotropic Material with Penalization) interpolation scheme [51] as follows:

$$\mathbf{k}(\rho) = \mathbf{k_0}\,\rho^p.$$

Here, $\mathbf{k_0}$ is the stiffness of the solid material, and $p = 3$ is the penalty factor, which is used for penalizing intermediate densities.

For manufacturing purposes, elements with a density below the specified threshold value were removed from the model, and retained elements with a density above the threshold value are considered parts.

There are two manufacturing methods for composite part production. The first method is related to cutting a part from a molded workpiece, which can, for example, be in the form of a rectangular plate. This case is named "constant molding", where material anisotropy exists but does not depend on the part shape. The second method is injection molding of the designed part. The second case is named "variable molding", where a coupled simulation of injection molding and structural optimization is required.

Unidisciplinary Topology Optimization Considering Constant Molding

Anisotropy was modeled by adjusting the elastic constants of the stiffness matrix. Since composites reinforced with 30–50 wt.% fiber along the flow direction exhibit double the stiffness compared to that across the flow direction [75], the anisotropy was incorporated by modifying stiffness matrix components. The mechanical characteristics of the anisotropic material were aligned with the global coordinate axis and remained constant throughout the TO, allowing for the optimization of structure arrangement based on a predefined stiffness distribution.

Multidisciplinary Topology Optimization: Considering Variable Molding

As explained in the previous section, modeling the anisotropy during structural analysis involves incorporating the relevant material characteristics of an anisotropic material. However, this approach is suitable for cutting structures from injection-molded plates (constant molding case) but not for molded structures (variable molding case). In molded structures, the fiber orientation within the channels, which are structural elements from the perspective of solid mechanics, cannot be defined until TO is performed.

The technique outlined in [75] was applied to account for fiber arrangement within the channels and dynamically adjust the mechanical characteristics of each structural element by recalculating the element's fiber orientation tensor. Algorithm 1 for multidisciplinary TO considering variable anisotropy is presented in Figure 2 and in the pseudocode. A detailed description of the algorithm implementation is available in [76].

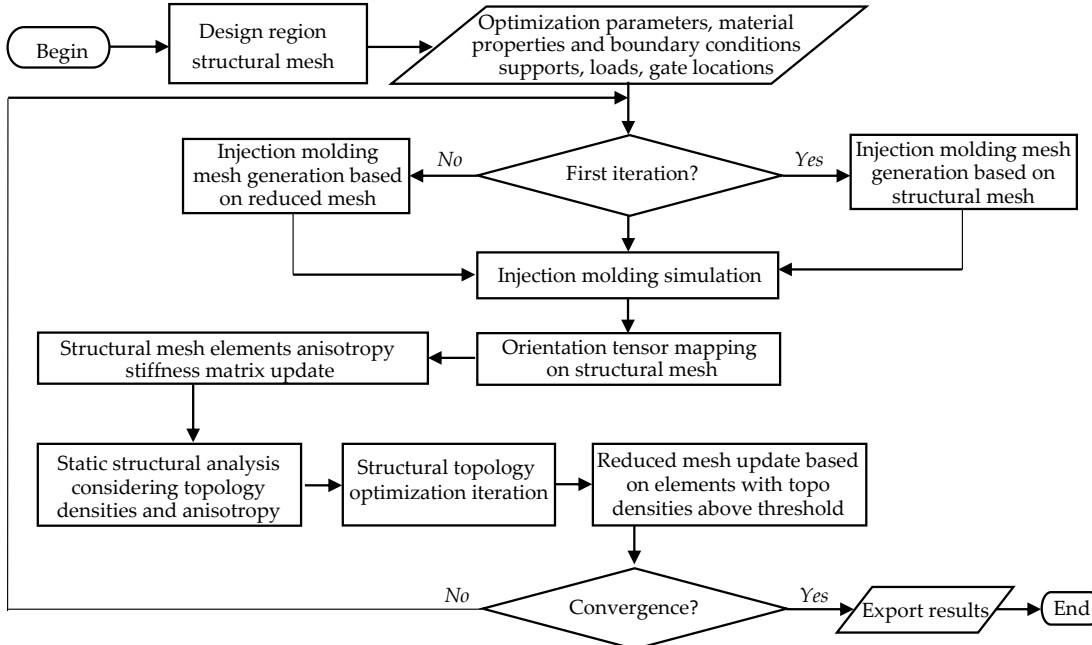

**Figure 2.** Multidisciplinary topology optimization considering a variable anisotropy flow chart.

The process begins with the introduction of initial values, including boundary conditions for structural analysis $BC_{struct}$ and injection molding simulation $BC_{injMold}$, material properties of the matrix, and fiber $MP_{matrix+fiber}$, injection molding material $MP_{injMold}$, TO parameters $OP_{topoOpt}$, topology density threshold *th*, design region volume percentage to retain $V_{ret}$, interpolation tolerance between the injection molding and structural analysis meshes $\delta_{map}$, and an optional *geo.stp* file for geometry. $MP_{matrix+fiber}$ and $OP_{topoOpt}$ are saved as materialProperties.txt and topoParameters.txt, respectively, for correct utilization in AnisoTopo [67]. The structural domain region mesh, elements participating in TO, and elements related to boundary conditions are obtained from Ansys Workbench and saved as designRegionMesh.ans, design.txt, and frozen.txt, respectively. TO with variable fiber orientation proceeds after specifying the convergence criterion for the TO algorithm, which is the relative difference $\varepsilon$ between the previous and current objective function values, not exceeding the objective relative difference $\varepsilon_{obj}$. Key moments in TO include:

First, the structural mesh is exported to Autodesk Moldflow. If it is the first iteration, reducedMesh.ans will be the same as designRegionMesh.ans, as it has not been previously reduced. Otherwise, the structural mesh is reduced using the "delete_elements" algorithm and saved as reducedMesh.ans (details in [76]). At the molding simulation stage, the mesh is refined, including the removal of areas not connected with injection locations. Second, the reducedMesh.ans, $BC_{injMold}$, and $MP_{injMold}$ are introduced to Moldflow for calculating

the fiber orientation tensor **A**, which is exported along with the injection molding mesh $\text{mesh}_{\text{injMold}}$ and saved in the files meshMoldFlow.pat and fiberOrientMoldFlow.xml, respectively. Fiber orientation mapping from injection molding to structural analysis mesh is performed in DigimatMAP, and the mapped fiber orientation **A′** is stored in the file fiberOrientAnsys.xml. The mapping step also extrapolates the orientation tensor field to regions not included in the molding simulation design domain regions, which allows the smooth material characteristics field at the part boundaries to be obtained.

Third, in AnisoTopo [67], **A′** is introduced to calculate the anisotropic stiffness matrix (K) using the Advani-Tucker orientation averaging technique applied to the material stiffness matrix [4]. AnisoTopo's code employs the Mori–Tanaka homogenization method to compute mechanical properties.

The composite strain and stress are contingent on the strain and stress of both the matrix and fiber in proportion to their volume fractions, as follows:

$$\varepsilon = \left(1 - \phi_f\right)\varepsilon_m + \phi_f\varepsilon_f,$$

$$\sigma = \left(1 - \phi_f\right)\sigma_m + \phi_f\sigma_f,$$

where $\phi_f$ is the fiber volume fraction, and the subscripts $m$ denote the strain and stress of matrix values, while the subscripts $f$ correspond to the strain and stress of the fiber.

The unidirectional short-fiber-reinforced composite material is modeled as transversely isotropic. The elastic moduli, as introduced by Tandon and Weng [77], were used to calculate the elastic coefficients:

$$\frac{E_{11}}{E_m} = \frac{1}{1 + \frac{\phi_f(A_1 + 2v_m A_2)}{A_6}}$$

$$\frac{E_{22}}{E_m} = \frac{1}{1 + \frac{\phi_f[-2v_m A_3 + (1 - v_m)A_4 + (1 + v_m)A_5 A_6]}{2A_6}},$$

where $E_m$ and $v_m$ are the Young's modulus and Poisson ratio of the matrix, respectively. The parameters $A_i$ are the functions of Eshelby's tensor and can be found in [78]. In this particular study, we employ Eshelby's tensor for an elliptical inclusion, which depends on the fiber's aspect ratio. Tucker's averaging procedure is used to account for the fiber orientation tensor, described as follows:

$$C_{ijkl} = B_1 a_{ijkl} + B_2\left(a_{ij}\delta_{kl} + \delta_{ij}a_{kl}\right) + B_3\left(a_{ik}\delta_{jl} + a_{il}\delta_{jk} + a_{jl}\delta_{ik} + a_{jk}\delta_{il}\right)$$
$$+ B_4\left(\delta_{ij}\delta_{kl}\right) + B_5\left(\delta_{ik}\delta_{jl} + \delta_{il}\delta_{jl}\right),$$

where $a_{ijkl}$ is the fourth-order fiber orientation tensor, $\delta_{ij}$ is the second-order unit tensor, and the coefficients $B$ are related to the components of the stiffness matrix of the transversely isotropic unidirectional composite [79]. This resulting tensor is linked to each mesh element and exported as apdl_pre.txt to Ansys Mechanical Workbench.

Fourth, the structural analysis is carried out. In all iterations but the first, the structural analysis is performed on the current topology as determined by the topology density $\rho$ (details in [75]). Subsequently, $\varepsilon$ is calculated, and the topology is extracted based on $V_{\text{def}}$ and exported as density.topo. If the current $\varepsilon$ is less than $\varepsilon_{\text{obj}}$ for the number of times specified by the k-iterations criterion $K_\varepsilon$ [80] (in this work, three times in a row), TO is stopped, and the last topology is exported as topoOptStruct.stl. Otherwise, the counter g is incremented by 1, and the loop repeats until convergence is achieved.

---

**Algorithm 1.** Multidisciplinary topology optimization.

---

**Input:** $BC_{struct}$, $BC_{injMold}$, $MP_{injMold}$, $MP_{matrix+fiber}$, $OP_{topoOpt}$, *th*, $\delta_{map}$, $V_{def}$, $\varepsilon_{obj}$, *geo.stp (Optional)*
**Output:** *topoOptStruct.stl*
**write** materialProperties.txt ← $MP_{fiber+matrix}$
**write** topoParameters.txt ← $OP_{topoOpt}$
$mesh_{struct}$, $design_{elements}$, $frozen_{elements}$ = AnsysWorkbench_Mesh($BC_{struct}$, *geo.stp*);
**write** designRegionMesh.ans ← $mesh_{struct}$, design.txt ← $design_{elements}$, frozen.txt ←
$frozen_{elements}$;
g = 1;
counter_epsilon = 0;
**while** (counter_epsilon < $K_\varepsilon$) **do**
    **if** g == 1 **then**
        reducedMesh.ans = designRegionMesh.ans;
    **else**
        domain_mesh_reduced = delete_elements(designRegionMesh.ans, *th*, (density.txt)$_{g-1}$);
        **write** reducedMesh.ans ← domain_mesh_reduced
    **end if**
    $mesh_{injMold}$, **A** = AutodeskMoldFlow(reducedMesh.ans, $BC_{injMold}$, $MP_{injMold}$);
    **write** meshMoldFlow.pat ← $mesh_{injMold}$, fiberOrientMoldFlow.xml ← **A**;
    **A′** = DigimatMAP(fiberOrientMoldFlow.xml, meshMoldFlow.pat, designRegionMesh.ans, δ);
    **write** fiberOrientAnsys.xml ← **A′**;
    $\mathbf{K_{EL}}$ = AnisoTopo(materialProperties.txt, fiberOrientAnsys.xml, topoPararmeters.txt,
designRegionMesh.ans, design.txt, frozen.txt);
    **write** apdl_pre.txt ← $\mathbf{K_{EL}}$;
    **if** g == 1 **then**
        $W_g$ = AnsysWorkbench_StructuralAnalysis(*designRegionMesh.ans*, apdl_pre.txt,
*$BC_{struct}$*);
    **else**
        $W_g$ = AnsysWorkbench_StructuralAnalysis(*designRegionMesh.ans*, apdl_pre.txt,
$BC_{struct}$, **ρ**);
        $\varepsilon_g$ = | $(W_g - W_{g-1})/W_{g-1}$ |
        **if** $\varepsilon_g$ <= $\varepsilon_{obj}$ **then**
            counter_epsilon ++
        **else**
            counter_epsilon = 0
        **end if**
    **end if**
    **ρ** = AnsysWorkbench_TopologyOptimization_Iteration(designRegionMesh.ans, design.txt,
frozen.txt, $V_{def}$, topoPara, apdl_pre.txt, $W_g$)
    **write** density.topo ← **ρ**;
    Convert density.topo to (density.txt)$_g$ with HDFView();
    g++
**end while**
topoOptStruct = delete_elements(designRegionMesh.ans, *th*, (density.txt)$_{g-1}$);
**write** *topoOptStruct.stl* ← topoOptStruct

---

### 2.2.2. Metrics for Evaluating the Structure Design Quality of Composite Materials

The load-carrying factor (LCF), denoted as G, along with its coefficient $C_K$, is employed to assess the quality of a structural arrangement [55,56]. Their typical formulations are defined as follows:

$$G = \int_V \sigma_{eq} dV, \tag{2}$$

$$C_K = \frac{G}{Fl} \tag{3}$$

Here, $\sigma_{eq}$ represents the equivalent stress, V is the volume of the structure, F is the characteristic load in N, and l is the characteristic linear dimension in m (l represents the distance between areas where loads are applied to the locations of the supports). To

properly evaluate the quality of the structural arrangement in composite materials, the LCF has been redefined based on stress criteria.

For structures made of isotropic materials, the LCF is expressed as follows:

$$G = \sigma^{UTS} \int_V F_{eq} dV, \tag{4}$$

Here, $\sigma^{UTS}$ is the ultimate tensile stress, and $F_{eq} = \sigma_V \,/\, \sigma^{UTS}$ is the maximum stress criterion, defined as the ratio of von Mises stress $\sigma_V$ to the material's ultimate tensile stress $\sigma^{UTS}$. The LCF coefficient remains the same as in Equation (3).

In the case of structures made of anisotropic materials, the LCF is expressed as follows:

$$G_{TH} = \sigma_0^{UTS} \int_V F_{TH} dV, \tag{5}$$

Here, $\sigma_0^{UTS}$ is the ultimate tensile strength either along the longitudinal direction (along the fiber), and $F_{TH}$ is the average Tsai–Hill criterion, determined using Advani-Tucker's averaging procedure [66]. It is defined as follows:

$$F_{TH} = D_1 a_{ijkl} + D_2\left(a_{ij}\delta_{kl} + a_{kl}\delta_{ij}\right) + D_3\left(a_{ik}\delta_{jl} + a_{il}\delta_{jk} + a_{jl}\delta_{ik} + a_{jk}\delta_{il}\right) + D_4\left(\delta_{ij}\delta_{kl}\right) + D_5\left(\delta_{ik}\delta_{jl} + \delta_{il}\delta_{jk}\right), \tag{6}$$

where $D_1 = F_{TH1111}^{ud} - 2\,F_{TH1122}^{ud} + F_{TH2233}^{ud} - 4\,F_{TH1212}^{ud} + F_{TH2323}^{ud}$; $D_2 = F_{TH1122}^{ud} - F_{TH2233}^{ud}$; $D_3 = F_{TH1212}^{ud} - F_{TH2323}^{ud}$; $D_4 = F_{TH2233}^{ud}$; and $D_5 = F_{TH2323}^{ud}$. The values of the Tsai–Hill criteria tensor are determined using the following expression:

$$F_{TH}^{ud} = \frac{\sigma_{11}^2}{X^2} - \frac{\sigma_{11}(\sigma_{22} + \sigma_{33})}{X^2} + \frac{\sigma_{22}^2 + \sigma_{33}^2}{Y^2} + \left(\frac{1}{X^2} - \frac{2}{Y^2}\right)\sigma_{22}\sigma_{33} + \frac{\sigma_{12}^2 + \sigma_{13}^2}{S^2} + \left(\frac{4}{Y^2} - \frac{1}{X^2}\right)\sigma_{23}^2, \tag{7}$$

Here, $\sigma_{ij}$ represents the components of the stress tensor (component 11 corresponds to the fiber's longitudinal axis, etc.), X is the longitudinal strength limit, Y is the transverse strength limit, and S is the transverse shear strength.

The LCF coefficient, accounting for the anisotropy, can be written as follows:

$$C_K^{TH} = \frac{G_{TH}}{Fl} \tag{8}$$

It should be mentioned that the failure criteria presented above are not considered in the topology optimization problem since the primary scope of the given work is to investigate the influence of the anisotropic properties of SFRP on classical topology optimization results aimed at obtaining a minimum compliance design.

The second metric that allows comparison of the stiffness of differently loaded structures made of different materials is the normalized specific stiffness $\bar{k}$, which is calculated as the ratio of $F/m$ to $\delta/l$ in the elastic zone:

$$\bar{k} = \frac{Fl}{m\delta}, \tag{9}$$

where F is the loading force, m is the bracket mass, $\delta$ is the displacement of the bracket lug along the line of force, and l is the distance from the supports to the line of force application.

### 2.2.3. Bracket Manufacturing and Load Testing Technique

The variable molding brackets and plates for cutting constant molding bracket production were manufactured by injection molding using a Negri Bossi VE 210-1700 injection molding machine. The filling parameters were as follows: melt temperature 230 °C, mold temperature 80 °C. The pellets were dried before the injection at a temperature of 90 °C for 4 h in a plastic pellet dryer. Mold for variable molding brackets injection molding was manufactured from St-3 steel plates underwent CNC machining using a 4-flute 4 mm AlTiN D4x50x4Dx4F coated carbide endmill.

A static test was conducted to verify the developed methodology on both brackets. Before conducting the static test, the brackets were weighed using an electronic balance with a resolution of 0.01 g. Both brackets are loaded along the Y-axis until they fail. Restricting plates were used to ensure that the brackets failed due to fracture and not buckling.

The mechanical testing equipment for both brackets consisted of an MTS 322 testing machine with mechanical grips, an MTS 793 controller, and an MTS 661.20F-03 force sensor, as well as special equipment shown in Figure 3, which included auxiliary rods and plates. Before performing the tensile test, the bracket support 1 and loading rod 2 were positioned. A mounting tool was used to check the collinearity between the bracket's support and loading rod axes, and a construction level was used to ensure the vertical straightness of the loading scheme. Subsequently, the bracket was positioned between the bracket support plates and secured with M6 bolts. Displacement-restricting plates 3 were attached to bracket support 1 using M6 bolts. Linking rod 4 is attached to loading rod 1 with M6 bolts. The need for using the Z-displacement restricting plates arises from the conducted linear buckling analysis. The alignment of the bracket and linking-rod axes was achieved by adjusting the height of the loading rod. Finally, the bracket and linking rod were fixed in place using M6 bolts. After positioning all the components and ensuring that the bracket was not pre-stressed, the gauge sensor was set to zero.

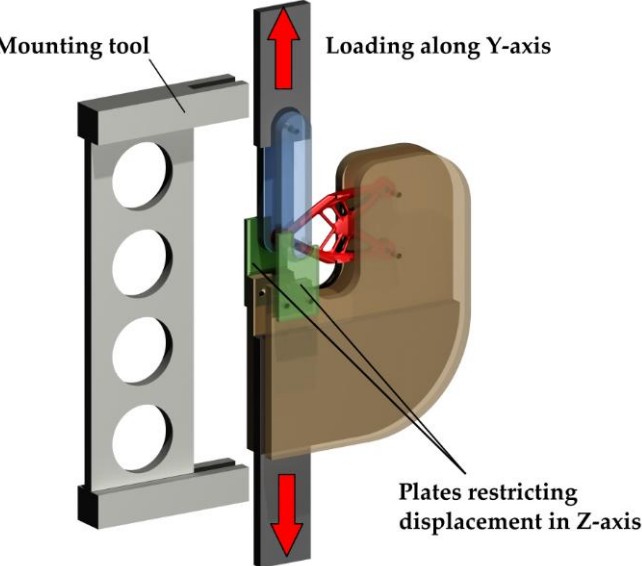

**Figure 3.** Loading scheme of special mechanical testing equipment for bracket testing.

## 3. Results

TO was performed on two case studies to obtain the structural layout for constant molding and variable molding cases. In the constant molding case, the topologies obtained with and without considering constant anisotropy were compared. In the variable molding case, the solutions obtained using fixed and variable fiber orientations were compared. The materials used were PA6 50GF, PA6 30CF, and D16T. The aluminum structures, along with the LCF coefficient, serve as a control for comparing the resultant topologies.

### 3.1. Topology of Optimal Constant Molding Structures

3.1.1. Topology Optimization and Structural Arrangement Quality Assessment

In the constant molding case, the design space is a rectangle with dimensions $75 \times 50 \times 4$ mm, as shown in Figure 4, where frozen elements near the loads and supports are marked in gray. The structured mesh consists of 47,862 elements, with an edge size of 0.25 mm. This was assessed by the element quality mesh metric, with a minimal value of 0.4009, a maximum value of 0.9949, an average value of 0.8834, and a standard deviation of $6.4556 \times 10^{-2}$.

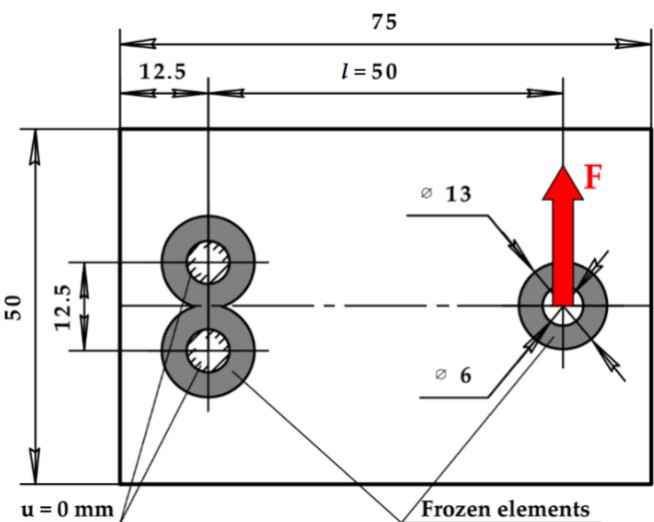

**Figure 4.** Topology optimization design region, boundary conditions, and load case.

Force F was applied to the elements located within the ring at the center-right side of the region, while the displacement of the elements within the rings on the left side was set to 0 mm. The design region volume percentage to retain $m_{ret}$ was set to 12%. The objective relative difference $\varepsilon_{obj}$ value was established as 0.1%. The material properties were defined as follows: for isotropic material, Young's modulus of 8 GPa and a Poisson's ratio of 0.25; for orthotropic material, the following elastic constants: $E_X$ = 13 GPa, $E_Y$ = 7 GPa, $E_Z$ = 6.5 GPa, $\upsilon_{XY}$ = 0.272, $\upsilon_{YZ}$ = 0.365, $\upsilon_{XZ}$ = 0.254, $G_{XY}$ = 1.979 GPa, $G_{YZ}$ = 1.639 GPa, and $G_{XZ}$ = 1.763 GPa (the X-axis corresponds to the bracket symmetry axis, the Y-axis corresponds to force direction, and the Z axis is determined by the right-hand rule).

During the topology optimization analysis, a bearing force is used as the loading boundary condition, and cylindrical supports are utilized as structural constraints. A linear static analysis was employed for the topology optimization.

The convergence results are plotted in Figure 5. A converged solution was achieved after 44 and 34 iterations for the TCA (topology optimized design with constant molding and anisotropic material) and TCI (topology optimized design with constant molding and isotropic material) cases, respectively. Here, "constant molding" means that a single injection molding simulation was performed before optimization, and the resulting solid material properties remain the same over the whole design domain. The resultant topologies are shown in Figure 6; the topology considering isotropy and constant anisotropy will be further referred to as TCI and TCA, respectively (where C stands for constant, even though isotropy is constant by definition).

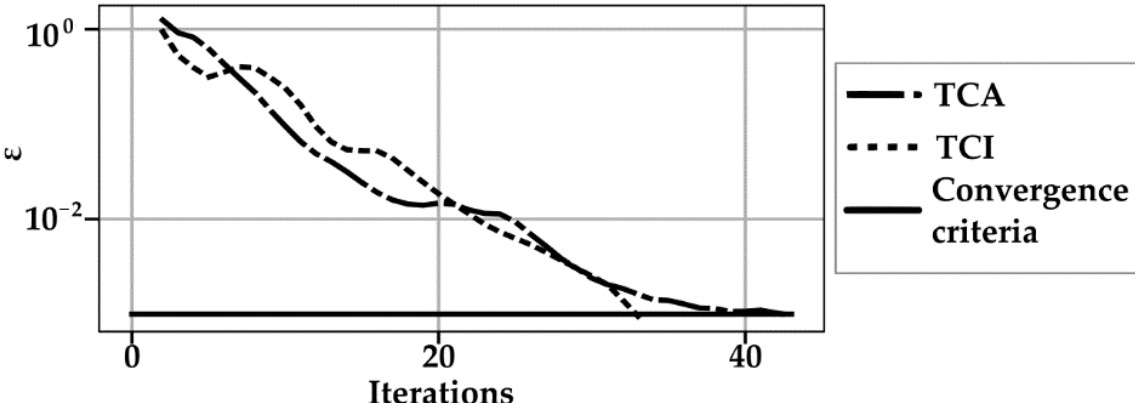

**Figure 5.** Topology optimization convergence plot by total strain energy relative error.

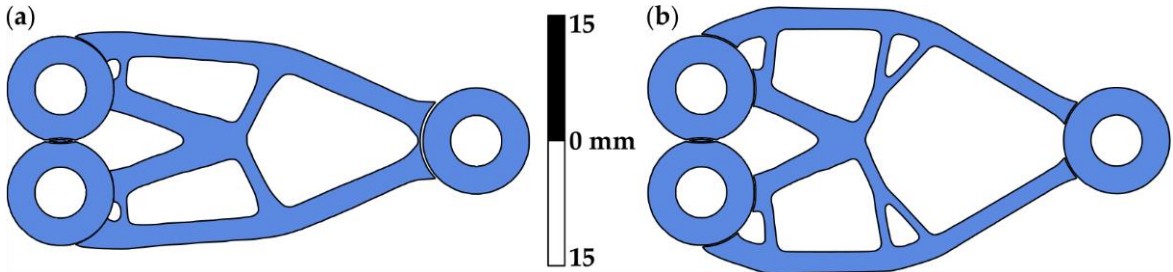

**Figure 6.** Resultant constant molding structures: (**a**) TCA and (**b**) TCI.

In the verification analysis, an anisotropic elasto-plastic material model is utilized, considering the orientation of reinforcing fibers. This model is calculated for the molded plate from which the parts are cut out. Loads and supports are applied to auxiliary cylindrical bodies, which are located inside holes and connected to the main part using nonlinear contact.

The von Mises-based and Tsai–Hill failure criterion fields of the resultant constant molding topologies, loaded at a force corresponding to the relation $^F/_m$ =70 N/gr, are presented in Figure 7. It can be observed that the von Mises failure criterion underestimates the strength of structural members loaded transversely in the fiber direction. Meanwhile, the Tsai–Hill failure criterion allows for a more accurate estimation of the strength of structures made of short-reinforced composite materials.

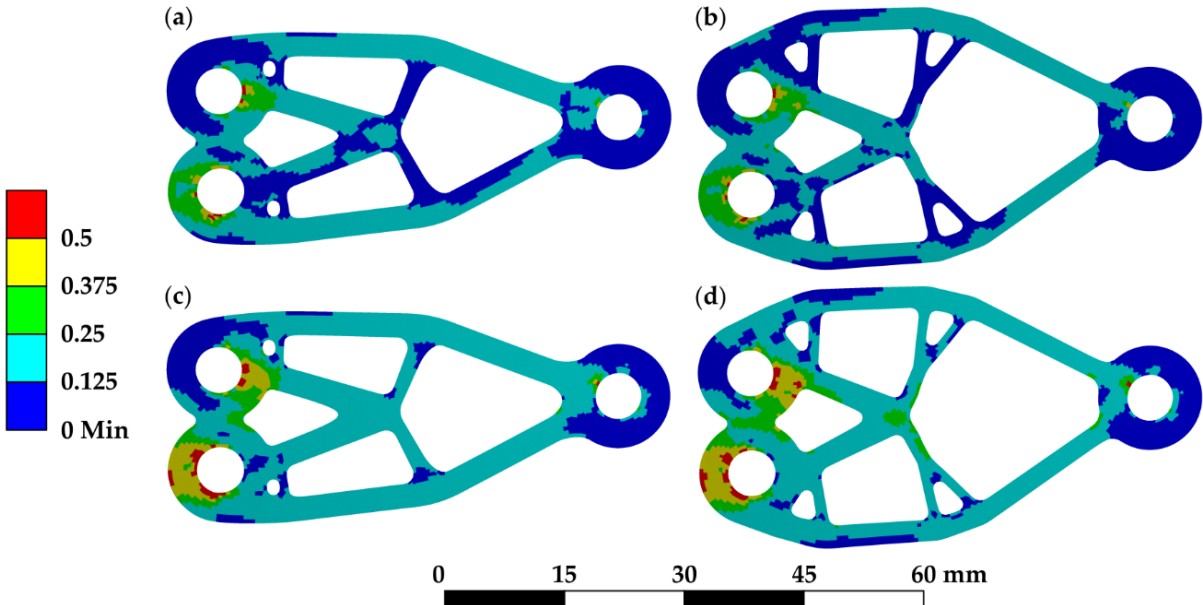

**Figure 7.** Failure criteria in the field of constant molding topologies: (**a**) von Mises-based failure criterion field of TCA; (**b**) von Mises-based failure criterion field of TCI; (**c**) Tsai–Hill failure criterion field of TCA; (**d**) Tsai–Hill failure criterion field of TCI.

The failure criteria and volume of each element in Equations (3) and (7) were multiplied and summed to calculate the LCF and LCF coefficients. The LCF and LCF coefficients of each topology and failure criterion combination are presented in Table 2. The relative percentage difference was calculated between the LCF coefficients of TCA and TCI.

**Table 2.** LCF coefficient baseline and reconstructed TCA and TCI.

| Topology | m, g | f, N | $C_K$ | $C_K{}^{TH}$ |
|---|---|---|---|---|
| | | PA6 50GF | | |
| TCA | 4.655 | 326.8 | 5.2928 | 5.6994 |
| TCI | 4.658 | 326.1 | 5.2482 | 5.8285 |
| | | PA6 30CF | | |
| TCA | 3.779 | 264.5 | 5.3407 | 6.7438 |
| TCI | 3.781 | 264.7 | 5.2874 | 7.1127 |
| | | D16T | | |
| TCA | 8.146 | 570.3 | 5.1964 | - |
| TCI | 8.152 | 570.6 | 5.2287 | - |

### 3.1.2. Influence of the Relationship between Elastic Moduli $E_1$ and $E_2$ of Composite Material on the Resulting Topology

An investigation of the influence of the relationship between $E_1$ and $E_2$ of composite material on the resulting part topology has been conducted, considering cases with two and four times higher anisotropy than those previously considered (refer to Figure 8). For each material, the normalized specific stiffnesses were calculated using $\bar{k}$ (9), representing the ratio of brackets from this material with TCA and TCI shapes. This ratio allows the evaluation of the potential for increasing the stiffness of the product by considering the material's anisotropy in the design.

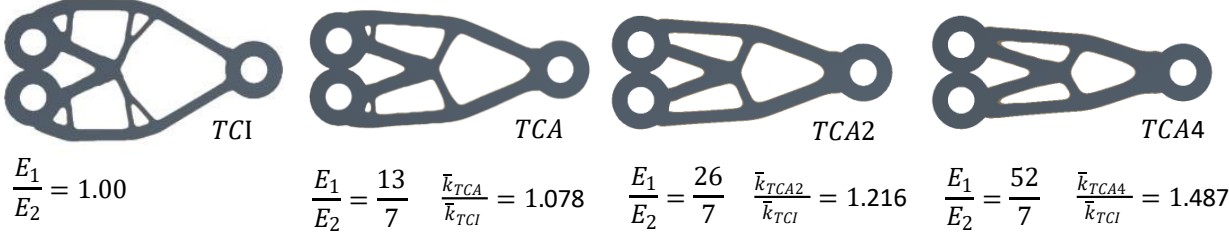

$$TCI \qquad TCA \qquad TCA2 \qquad TCA4$$

$$\frac{E_1}{E_2} = 1.00 \qquad \frac{E_1}{E_2} = \frac{13}{7} \quad \frac{\bar{k}_{TCA}}{\bar{k}_{TCI}} = 1.078 \qquad \frac{E_1}{E_2} = \frac{26}{7} \quad \frac{\bar{k}_{TCA2}}{\bar{k}_{TCI}} = 1.216 \qquad \frac{E_1}{E_2} = \frac{52}{7} \quad \frac{\bar{k}_{TCA4}}{\bar{k}_{TCI}} = 1.487$$

**Figure 8.** Influence of the relationship between $E_1$ and $E_2$ of composite material on the resulting topology.

It can be observed that as the material's anisotropy increases, the load-carrying elements of the structure align along the direction of maximum stiffness of the material, resulting in a decrease in the structure height and an increased importance of considering the material's anisotropy. Increasing the $E_1$ to $E_2$ ratio to 7.4 enables a 1.49-fold increase in the stiffness of the structure when considering the material's anisotropy during the topology optimization process. For further experimental verification, the case with an $E_1/E_2 = 13/7$ is chosen, which corresponds to the actual properties of the available composite material.

### 3.1.3. Experimental Verification

The TCA and TCI brackets (30 samples in total) were cut using a milling machine from plates molded from short, reinforced composite materials PA6 50GF, PA6 30CF, and, for comparison with isotropic material, from an aluminum D16T plate. Mechanical tests of the brackets were carried out on the MTS 322 machine (Figure 9). The linear buckling analysis shows that adding the Z-displacement restricting plates increases the load multiplier from 0.8 to 1.3 for plastic parts and prevents them from buckling. The results are presented in Figure 10 in terms of specific force and normalized deformation by the characteristic dimension (see Section 2.2.2).

Non-linear structural analysis was performed on the obtained topologies to compare the numerical and experimental results (Figure 11). The experimental lines are summarized in the form of average values over the samples, and the scatter field is determined by the value of the standard deviation. The normalized specific stiffness of the numerical

results was calculated from 25 to 50% of the maximal specific force for each TCA, TCI, and material combination. For experimental results, it was calculated from 2.07 to 4.12% for PA6 50GF, from 4 to 5.74% for PA6 30CF, and from 1.5 to 3% for D16T. The difference between experimental and numerical results is due to overestimating the stiffness of bolted joints and molded parts in the lug area and a rough approximation of testing tool flexibility. The integral characteristics of each structure as well as their normalized specific stiffness are presented in Table 3.

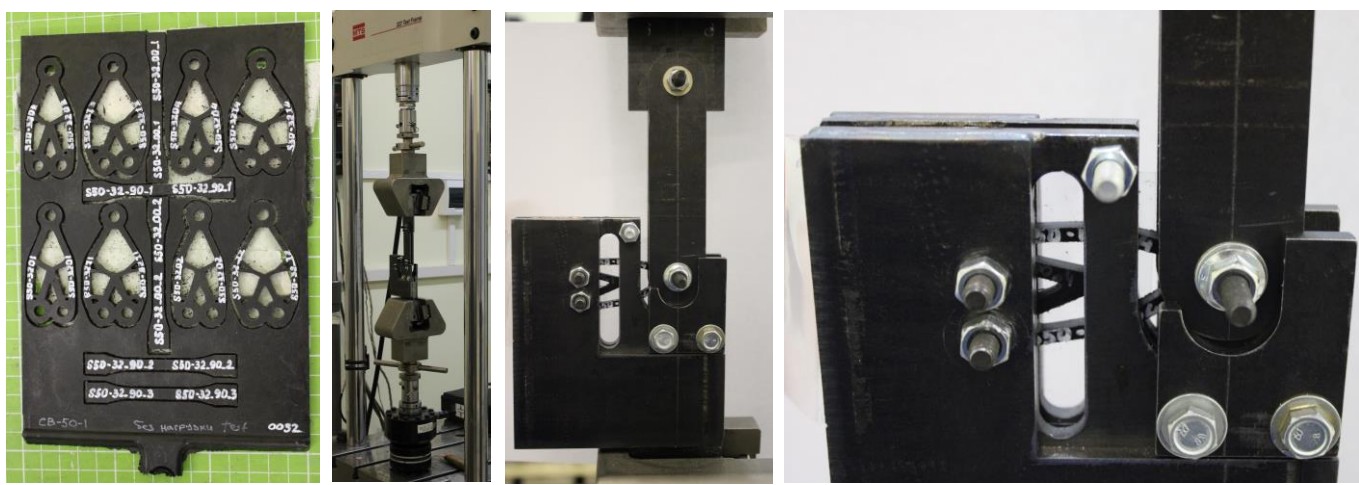

**Figure 9.** TCA and TCI bracket loading experiments.

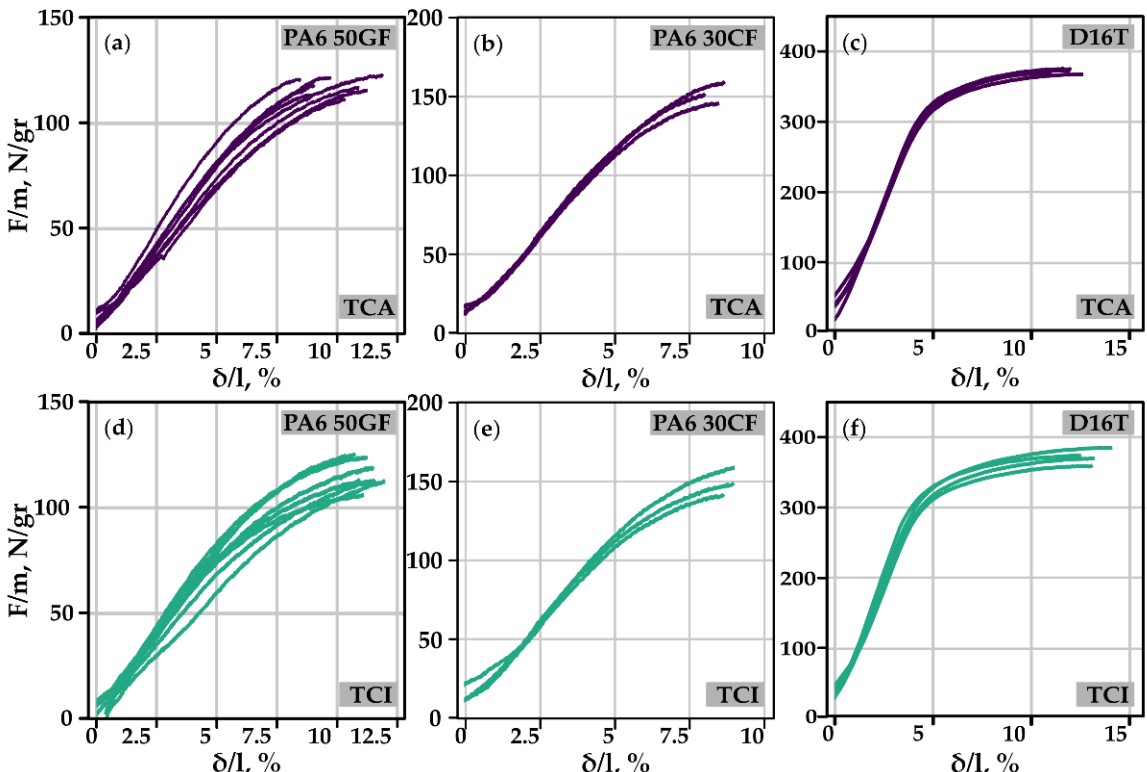

**Figure 10.** TCA and TCI experimental loading curves: (**a**) TCA made in PA6 50GF, (**b**) TCA made in PA6 30CF, (**c**) TCA made in D16T, (**d**) TCI made in PA6 50GF, (**e**) TCI made in PA6 30CF, (**f**) TCI made in D16T.

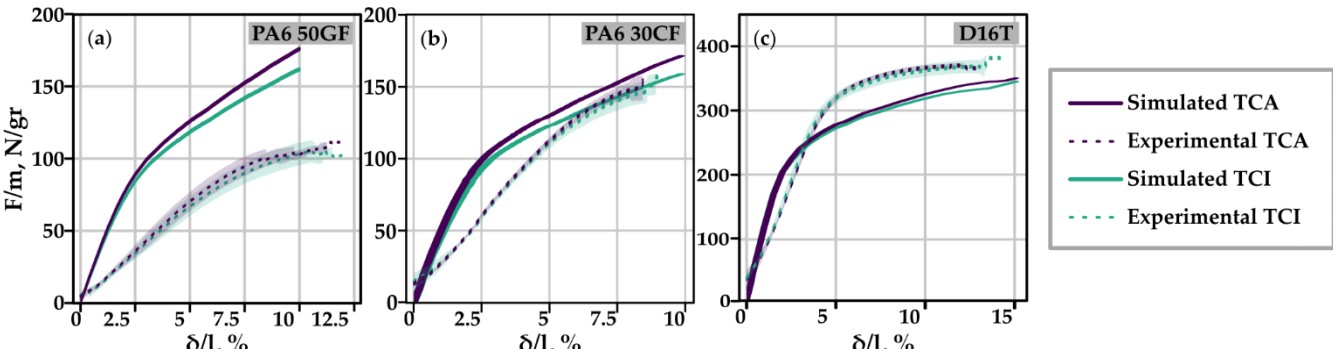

**Figure 11.** Specific force-normalized deformation curves of numerical and experimental TCA and TCI made of: (**a**) PA6 50GF; (**b**) PA6 30CF; and (**c**) D16T.

**Table 3.** Normalized specific stiffness of TCA and TCI.

| Topology | Normalized Specific Stiffness, N/gr | | Percentage Change from TCA to TCI, % |
|---|---|---|---|
| | TCA | TCI | |
| PA6 50GF | 3883 | 3661 | 6.06 |
| PA6 30CF | 4621 | 4194 | 10.18 |
| D16T | 11,474 | 11,654 | −1.54 |

The statistics of the experimentally obtained specific stiffness of TCA and TCI are shown in Table 4. The high coefficient of variation of D16T topologies results from the machining precision during the manufacturing of these topologies.

**Table 4.** Statistics of normalized specific stiffness of TCA and TCI.

| Material | Average, N/gr | | Standard Deviation, N/gr | | Coefficient of Variation, % | | Percentage Change from TCA to TCI, % |
|---|---|---|---|---|---|---|---|
| | TVA | TVI | TVA | TVI | TVA | TVI | |
| PA6 50GF | 1551 | 1479 | 140 | 167 | 9.03 | 11.26 | 4.87 |
| PA6 30CF | 1833 | 1737 | 52 | 96 | 2.84 | 5.54 | 5.53 |
| D16T | 7192 | 7584 | 76 | 406 | 1.06 | 5.36 | −5.17 |

### 3.2. Topology-Optimal Variable Molding Structures

3.2.1. Topology Optimization and Topology Assessment

In Figure 12, the design region and boundary conditions for the TO are presented. The design region has dimensions of $105 \times 60 \times 10$ mm. The optimization was carried out on a mesh of 163,325 tetragonal elements, each with a size of 1.5 mm. Force F was applied to the center-right side element of the region, while the elements located at the left-side corners had a fixed displacement of 0 mm. The target volume fraction to be retained was set to 20%, and the minimum size of the structural elements was 6 mm. Anisotropic materials PA6 50GF and PA6 30CF, as well as isotropic material D16T, were considered for the analysis.

As in the previous case of study, the resultant topology, considering anisotropy and isotropy, will be further referred to as TVA (topology optimized design with variable molding and anisotropic material) and TVI (topology optimized design with variable molding and isotropic material), respectively. Here, "variable molding" means that the injection molding simulation is performed at each optimization cycle, and solid material properties are updated according to orientation tensor field redistribution.

During the topology optimization stage, linear static analysis is performed. Fixed supports are utilized as structural constraints, and a uniform load is directly applied to the nodes of frozen elements.

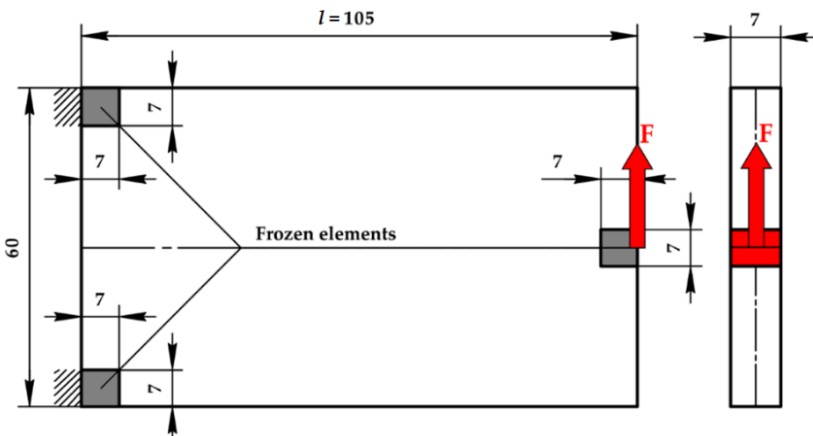

**Figure 12.** Topology optimization design region for the variable molding case.

The convergence progress is illustrated in Figure 13, where a converged solution was achieved after 71 and 66 iterations, respectively. The resultant topologies are displayed in Figure 14. A comparison of the results in Figure 14 reveals that considering the material's anisotropy in topology optimization results in changes to the structural layout. In Figure 14b, the connections of the main structural members become more rounded, and some rods are added in the central area of the part.

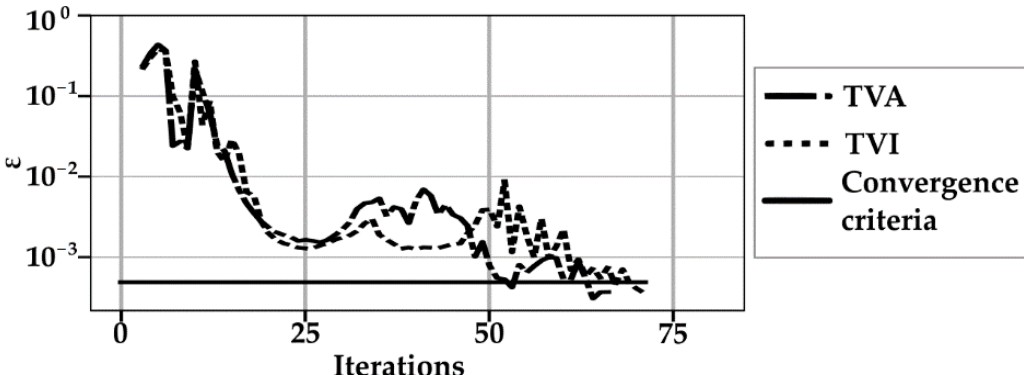

**Figure 13.** Topology optimization convergence plot by total strain energy relative error.

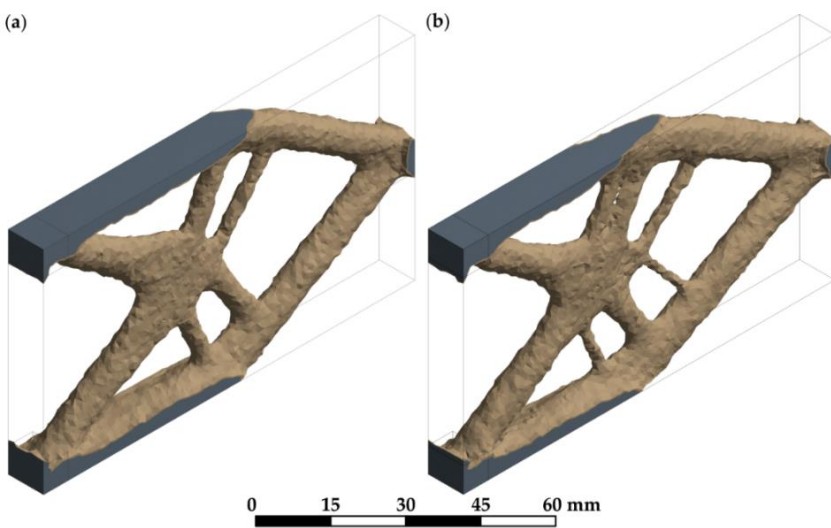

**Figure 14.** Obtained topologically optimal variable molding structures by using (**a**) fixed fiber orientation and (**b**) variable fiber orientation.

### 3.2.2. Topology Reconstruction

The resultant TO topologies from Section 3.2.1 are idealized results and are not final geometries ready for verification and manufacturing. They were manually reconstructed in Siemens NX. Figure 15 presents the baseline and reconstructed TVA and TVI topologies.

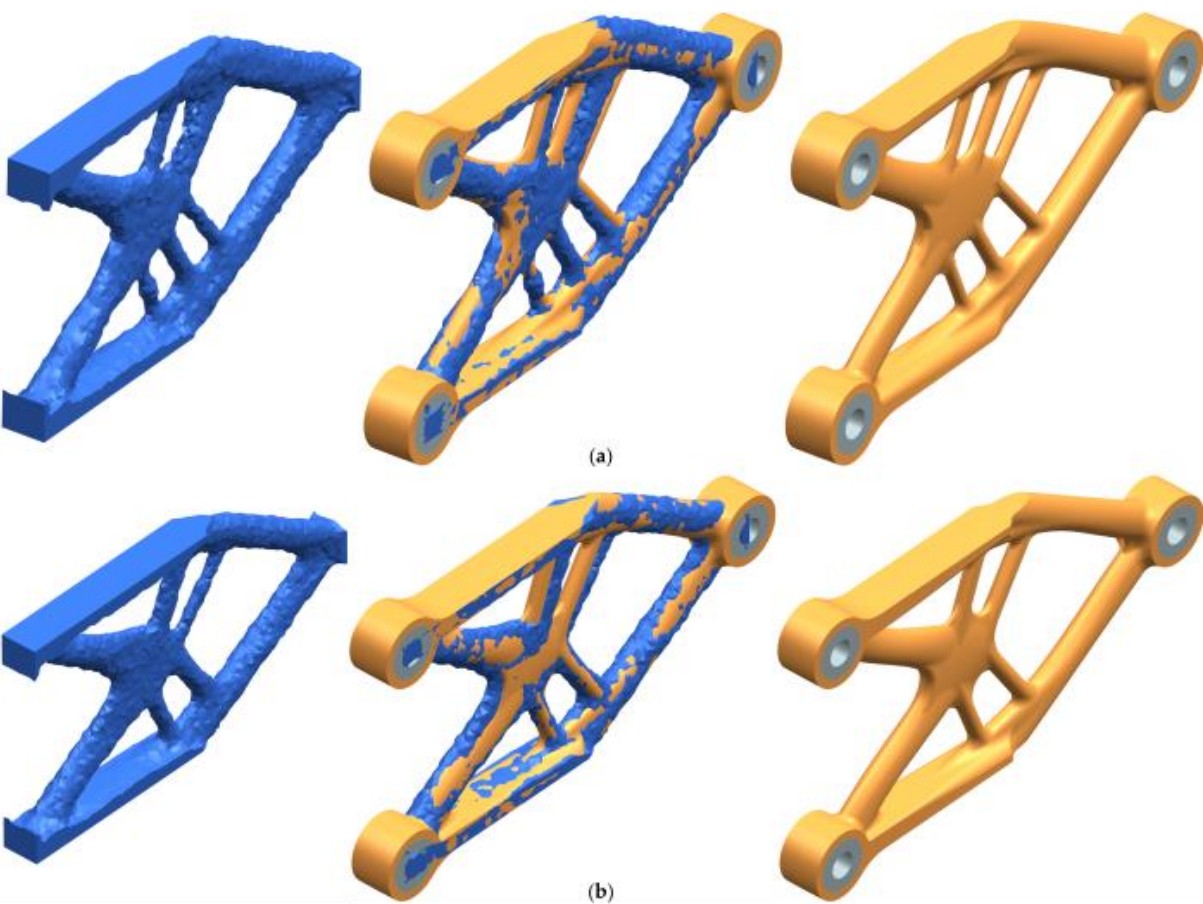

**Figure 15.** Topology reconstruction divided into three stages: the result of topology optimization (baseline), baseline and reconstructed topology overlaying, and reconstructed geometry: (**a**) TVA and (**b**) TVI.

Verification was conducted through nonlinear analysis using an anisotropic elasto-plastic material model, considering injection molding simulation. For both internal-exported and baseline designs, the loads and supports matched those used in the topology optimization. In the reconstructed cases, the loads and supports were applied to auxiliary cylindrical bodies modeled within holes and connected to the main part using nonlinear contact. Aluminum bushings were connected to the plastic parts via linear-bonded contact. Mesh sensitivity analysis revealed that the molding simulation yielded nearly identical results with both coarsely reduced meshes and fine meshes built on CAD models [81]. A comparison of the fiber orientation tensor at three points within the topology and evaluating the LCF coefficient of the topologies in different reconstruction stages were evaluated. Since injection molding simulation during TO was performed on a tetrahedral mesh, this mesh was evaluated as well. The internal-exported, baseline, and reconstructed topologies where the fiber orientation tensor was extracted and the components of their tensors are presented in Figure 16.

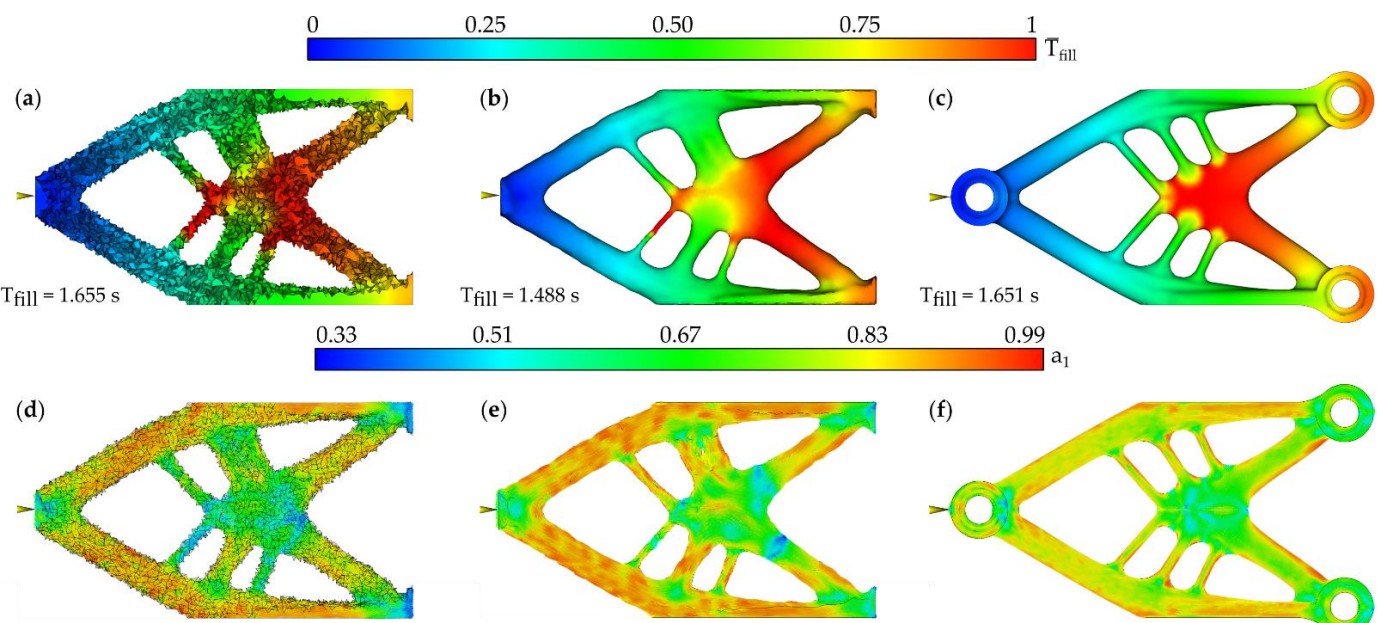

**Figure 16.** Flow fields: (**a**) internal-exported TVA; (**b**) baseline TVA; (**c**) reconstructed TVA; and (**b**) reconstructed TVI. Fiber orientation tensors: (**d**) internal-exported TVA; (**e**) baseline TVA; and (**f**) reconstructed TVA.

The internal-exported, baseline, and reconstructed topologies were subjected to lineal structural analysis using linear PA6 50GF, PA6 30CF, and D16T. Figure 17 illustrates the von Mises-based and Tsai–Hill failure criteria fields for the baseline, smoothed TVA and reconstructed TVA, and TVI when the topologies were subjected to loads corresponding to $^F/_M$ = 70 N/gr. Table 5 displays the corresponding $C_K$ values of both topologies and provides a comparison between them.

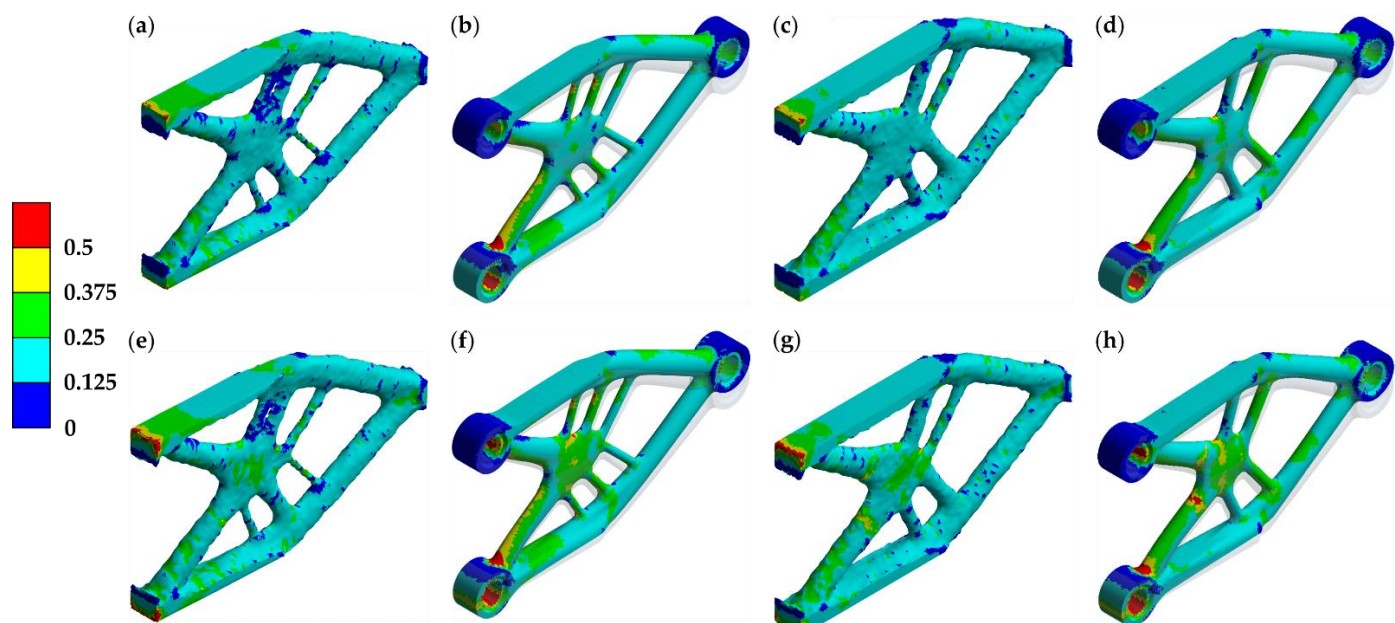

**Figure 17.** Tsai–Hill failure criterion fields: (**a**) baseline TVA; (**b**) reconstructed TVA; (**c**) baseline TVI; and (**d**) reconstructed TVI. Equivalent stress (von Mises-based) failure criterion fields: (**e**) baseline TVA; (**f**) reconstructed TVA; (**g**) baseline TVI; and (**h**) reconstructed TVI.

**Table 5.** LCF coefficient of internal-exported, baseline, and reconstructed TVA and TVI.

| Topology | Baseline | | | | Reconstructed | | | | Percentage Difference between Baseline and Reconstructed | |
|---|---|---|---|---|---|---|---|---|---|---|
| | m, g | f, N | $C_K^{eq}$ | $C_K^{TH}$ | m, g | f, N | $C_K^{eq}$ | $C_K^{TH}$ | $C_K^{eq}$ | $C_K^{TH}$ |
| | | | | | PA6 50GF | | | | | |
| TVA | 22.94 | 1606 | 3.3183 | 3.4341 | 25.59 | 1791 | 3.5230 | 3.6541 | 5.98% | 6.21% |
| TVI | 23.24 | 1627 | 3.3214 | 3.4437 | 26.17 | 1832 | 3.5481 | 3.6926 | 6.60% | 6.98% |
| | | | | | PA6 30CF | | | | | |
| TVA | 18.62 | 1304 | 3.3225 | 3.6881 | 20.76 | 1448 | 3.5273 | 3.9025 | 5.98% | 5.65% |
| TVI | 18.86 | 1320 | 3.3238 | 3.7103 | 21.24 | 1481 | 3.5539 | 3.9799 | 6.69% | 7.01% |
| | | | | | D16T | | | | | |
| TVA | 40.15 | 2810 | 3.3122 | - | 47.34 | 3314 | 3.5707 | - | 7.51% | - |
| TVI | 40.66 | 2846 | 3.3176 | - | 48.36 | 3386 | 3.5892 | - | 7.86% | - |

The failure criteria F and volume dV of each element, as described in Equations (3) and (7), were multiplied and summed to calculate the LCF coefficient. The LCF coefficient of each topology and material combination is presented in Table 5. The characteristic linear dimension l for the baseline topologies is 105 mm, whereas for the reconstructed topologies, it is 102.55 mm. The ultimate tensile stress $\sigma^{UTS}$ for PA6 30CF topologies is 169.35 MPa, while for D16T topologies, it is 476 MPa. The percentage change is calculated with respect to the correspondent baseline topology.

### 3.2.3. Experimental Validation

TVA and TVI brackets are made by injection molding (Figure 18). To strengthen the lugs, aluminum-embedded elements (bushings) were added. The simulation of the injection molding process was qualitatively validated by comparing the real and simulated flow at a specific filling time (Figure 19). The used design-stage flow model makes it possible to predict the material distribution during the molding process with good accuracy.

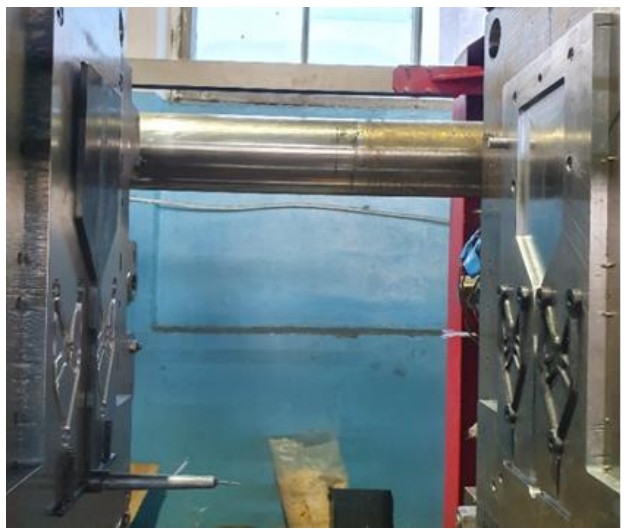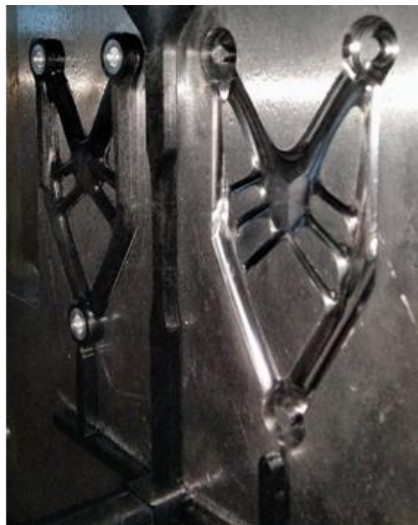

**Figure 18.** Manufacturing process of TVA and TVI brackets on an injection molding machine.

Fiber orientation was assessed by comparing the numerical fiber orientation against the real fiber orientation, which was observed under an electronic microscope, Tescan Vega 3T, at the fracture zone. Figure 20 shows the points at which the fiber orientation was evaluated.

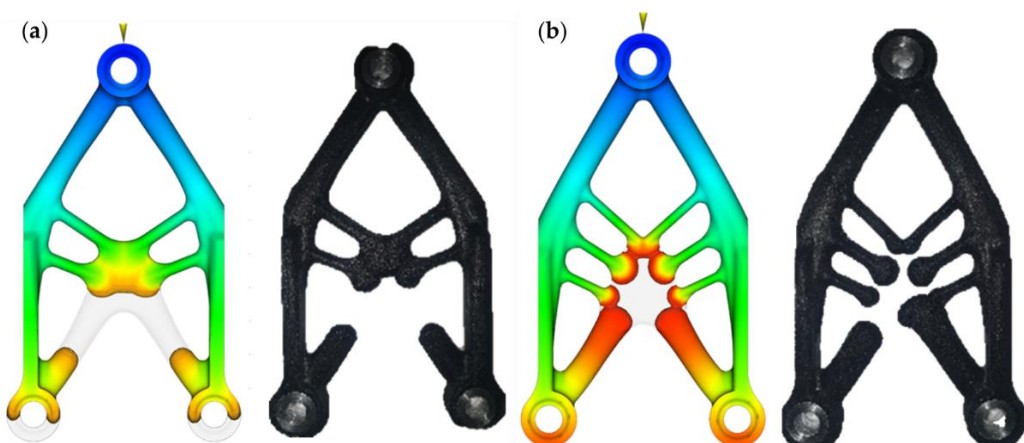

**Figure 19.** Flow fields: (**a**) simulated and experimental flow of TVA; and (**b**) simulated and experimental flow of TVI.

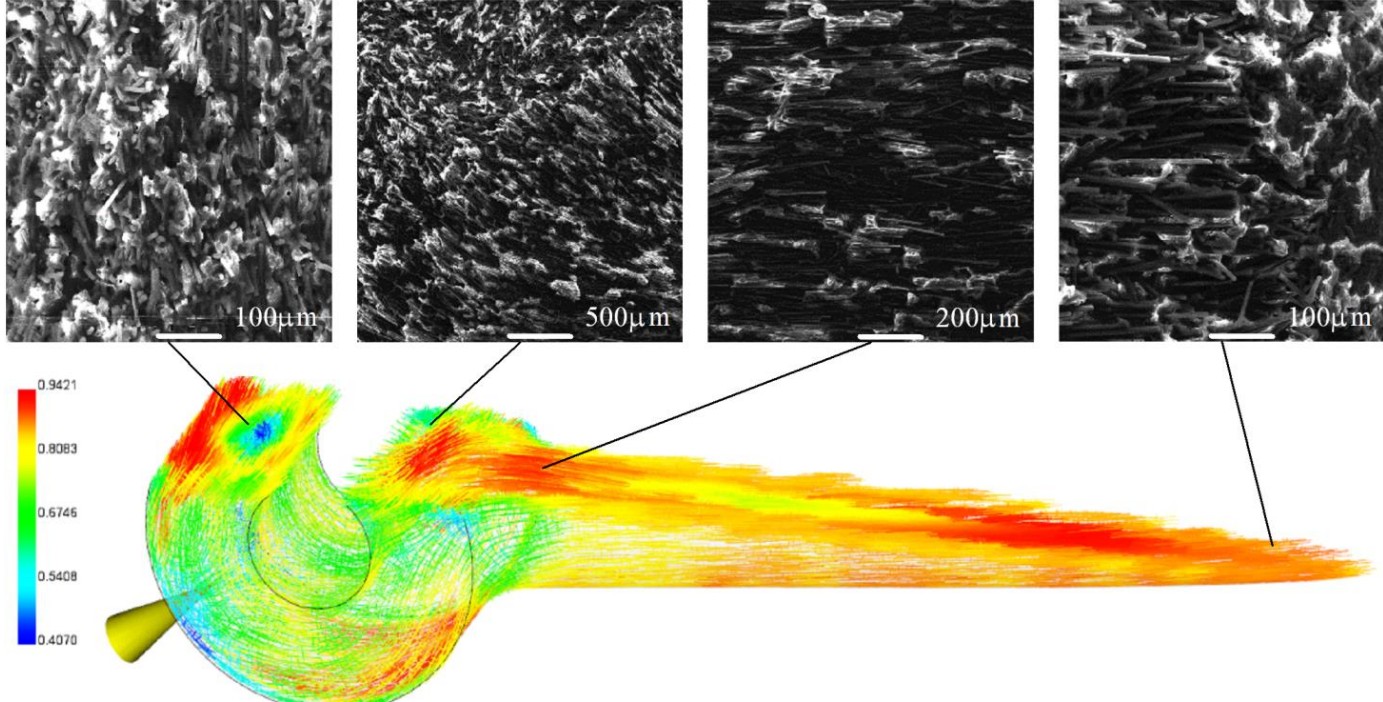

**Figure 20.** The fiber orientation at different fracture zones.

Mechanical tests of the TVA and TVI brackets from PA6 50GF, PA6 30CF, and D16T (51 samples in total) were carried out on the MTS 322 machine (Figure 21). The linear buckling analysis demonstrated that incorporating Z-displacement restricting plates enables an increase in the load multiplier from 0.52 to 1.3 for plastic parts, thereby preventing buckling. The experimentally obtained specific force-normalized deformation curves for TVA and TVI brackets are presented in Figure 22.

A non-linear structural analysis of TVA and TVI brackets was performed on the reconstructed topologies to compare the numerical and experimental results (Figure 23). The experimental lines are summarized in the form of average values over the samples and the scatter field by the value of the standard deviation.

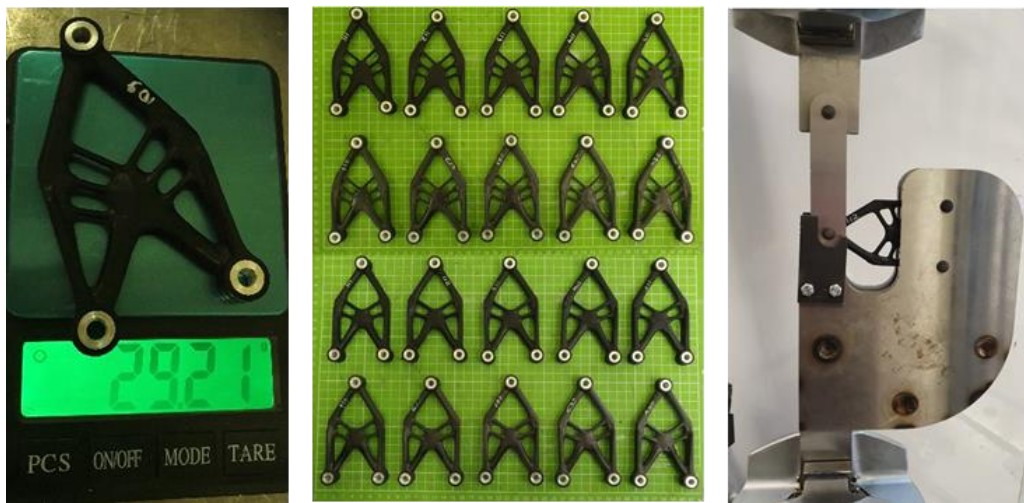

**Figure 21.** TVA and TVI bracket testing.

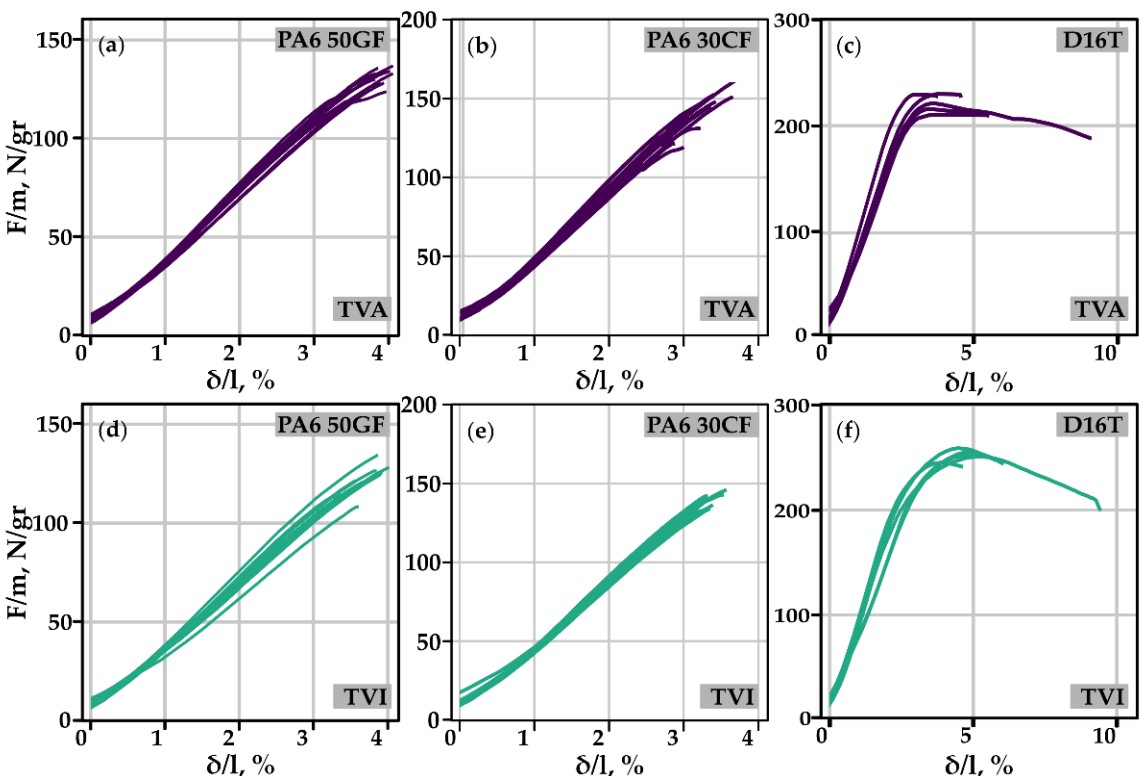

**Figure 22.** TVA and TVI bracket-loading experiments: (**a**) TVA made in PA6 50GF, (**b**) TVA made in PA6 30CF, (**c**) TVA made in D16T, (**d**) TVI made in PA6 50GF, (**e**) TVI made in PA6 30CF, (**f**) TVI made in D16T.

The normalized specific stiffness was calculated in the elastic zone in all cases with respect to normalized deformation. For the numerical results, the normalized specific stiffness ranged from 5.66 to 13.17%, while for the experimental results, it ranged from 7.5 to 15.8% for PA6 50GF and from 8.5 to 17.3% for PA5 30CF and D16T. The integral characteristics of each structure, as well as their normalized specific stiffness, are presented in Table 6.

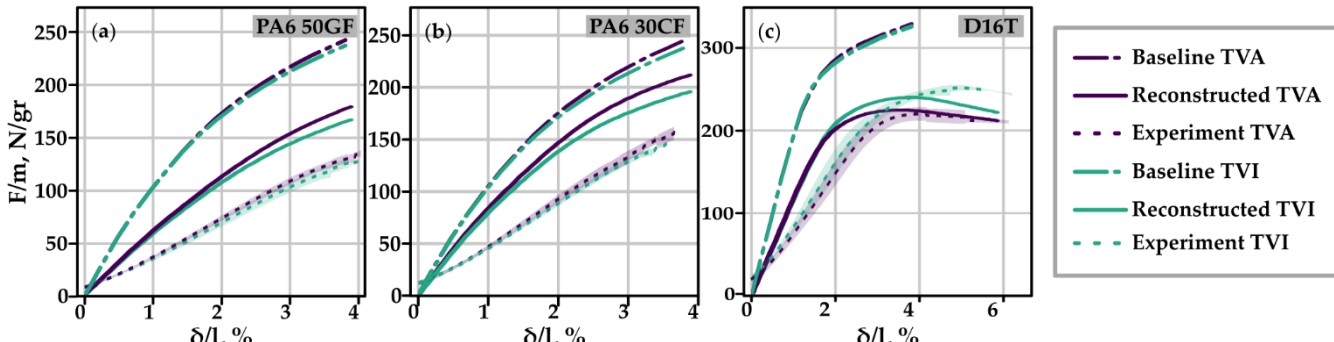

**Figure 23.** Specific force-normalized deformation curves of numerical and experimental TVA and TVI made of: (**a**) PA6 50GF; (**b**) PA6 30CF; and (**c**) D16T.

**Table 6.** Normalized specific stiffness and mass characteristics of the baseline and reconstructed TVA and TVI.

| Material | Topology | Normalized Specific Stiffness, N/gr | | Percentage Change from TVA to TVI, % |
|---|---|---|---|---|
| | | TVA | TVI | |
| PA6 50GF | Baseline | 7250 | 7187 | 0.88 |
| | Reconstructed | 5500 | 5260 | 4.56 |
| PA6 30CF | Baseline | 8993 | 8924 | 0.77 |
| | Reconstructed | 7209 | 6893 | 4.58 |
| D16T | Baseline | 19,737 | 19,869 | −0.66 |
| | Reconstructed | 11,838 | 11,613 | 1.92 |

The statistics of the experimentally obtained specific stiffness for TVA and TVI are shown in Table 7. The high coefficient of variation of D16T topologies is attributed to the machining precision during the manufacturing of these topologies.

**Table 7.** Statistics of the normalized specific stiffness of reconstructed TVA and TVI.

| Material | Average, N/gr | | Standard Deviation, N/gr | | Coefficient of Variation, % | | Percentage Change from TVA to TVI, % |
|---|---|---|---|---|---|---|---|
| | TVA | TVI | TVA | TVI | TVA | TVI | |
| PA6 50GF | 3529 | 3289 | 179 | 267 | 5.09 | 8.13 | 7.30 |
| PA6 30CF | 4533 | 4290 | 216 | 187 | 4.77 | 4.36 | 5.66 |
| D16T | 7293 | 7875 | 775 | 1150 | 10.63 | 14.60 | −7.39 |

## 4. Discussion

Assessment of the topologies by the LCF coefficient (see Sections 3.1.1 and 3.1.2) reveals that the classic coefficient $C_K$ is more suitable for structures made of isotropic materials. The classic $C_K$ does not convey any information regarding anisotropy, making it incapable of accurately estimating the quality of structures made from anisotropic material. For example, the classic $C_K$ indicates the advantage of TCI made from anisotropic materials over TCA, despite the latter having lower specific stiffness (see Tables 2 and 3). The $C_K^{TH}$ incorporates the effect of anisotropy on the stress state of the structure by optimizing the placement of structural elements to achieve more efficient mold filling. The minimal $C_K^{TH}$ values of the TCA and TCI correspond to the maximal specific stiffness of these topologies. However, a significant drawback arises from the fact that the integral over the volume of the failure criteria is the product of the average criteria values over the volume, making

LCF insensitive to both under- and over-stressed elements. We recommend evaluating the structure's topology not only using any of the possible formulations of the LCF coefficient but also employing a metric based on the coefficient of variance of the failure criteria (where the average and deviation are calculated from the failure criteria value over the volume). Moreover, reformulating the objective function to minimize both the average and deviation of the total strain energy of the topology should lead to a more equally strong structure.

The results obtained in Section 3 confirm that considering anisotropy during TO increases the normalized specific stiffness of the resultant topologies. Specifically, the normalized specific stiffness increased numerically by 6.06–10.18% and 0.77–0.88% in the constant and variable molding cases of study, respectively, for the available composite materials (see Sections 3.1.3 and 3.2.3). Increasing the $E_1$ to $E_2$ ratio to 7.4 enables a 49% increase in the stiffness of the structure in a constant molding case, which shows the possibility of increasing the effect in the presence of materials with greater anisotropy. In this work, all resulting topologies correspond to truss structures, which consist mostly of rods. In the variable molding case, all rods have almost the same elastic properties due to the alignment of fibers along their axes during injection molding. The impact of considering anisotropy during TO is more pronounced in the constant molding case. It can be assumed that the contrast in the normalized specific stiffness between the constant and variable molding cases is due to the regular orientation of the flow—and the fibers within it—along the structural elements in the variable molding case, causing the mechanical characteristics in these elements to be closer to those of the $0°$-oriented material. In the case of thin-walled structures, the effect of increasing stiffness, considering variable molding anisotropy, can be more prominent.

The increase in the change in the normalized specific stiffness of the reconstructed topologies compared to the baseline topologies (see Section 3.2.3) can be attributed to inaccuracies introduced by the CAD engineer during reconstruction and the replacement of fixed supports with bolted joints. Moreover, an increment is observed when comparing the $C_K^{TH}$ of the baseline and reconstructed topologies, suggesting that $C_K^{TH}$ and the normalized specific stiffness are correlated and that $C_K^{TH}$ serves as an indicator of the degree of pristine (flawless) structure compared to the baseline topology. The experimental results showed that the normalized specific stiffness increased by 4.84–5.63% and 5.66–7.30% in the constant and variable molding cases, respectively (see Sections 3.1.3 and 3.2.3). This indicates that two important points in this work—making the stiffness matrix dependent on the fiber orientation tensor and obtaining the fiber orientation tensor by solving the molding equations along with the Folgar–Tucker's continuity equation—allow us to achieve a stiffer structure (since the 2D case does not account for these points). The deviation of the numerical results from the experimental results can be attributed to differences in the loading scheme, the material model, and the omission of the weld lines during the numerical calculation. During the experiment with TVA and TVI brackets, it was noted that the poor adhesion between the aluminum bushing and the SFRP structure generated a weak joint interfacial strength, which could be the reason for the lower normalized specific stiffness than that predicted by the numerical model. Therefore, the investigation of the adhesion of aluminum and titanium alloys to SFRP can be part of future studies.

Regarding the topology, it is evident that TO attempts to orient the structural elements along the fiber direction when accounting for anisotropy, in contrast to TO in an isotropic medium, where the elements are positioned further from the symmetry line to reduce bending moments. The same topology distribution can be observed in the results of the work [61], where fiber orientation was simultaneously and sequentially considered. Therefore, it is confirmed that for better structural performance of the structural elements, they should be oriented in the direction of the fiber, in other words, in the direction of maximum stiffness. Moreover, this last statement also demonstrates the minimal difference between the topologies in the variable molding case.

We have several recommendations for improving the repeatability of this work. First, the maximum strain rate at which the topologies are loaded during the tensile test should

be 0.01 s$^{-1}$. Second, the effect of boundary conditions should be minimized to achieve a higher contrast between solutions; alternatively, different boundary conditions should be investigated. For instance, increasing the design region, placing the loads further from the support, or analyzing complex loading schemes such as the geometry presented in the work [82]. Third, changing the type of material model used during TO from linear to non-linear should be considered, as previous works [83] have demonstrated that this leads to stronger structures.

## 5. Conclusions

In this work, a methodology for obtaining topology-optimal structures made of short fiber-reinforced composites (SFRC) while considering the material's anisotropy was presented. The calculation of the composite material's stiffness matrix was performed using the fiber orientation tensor, which was obtained by solving the plastic molding equations for non-Newtonian fluids. Both calculations and experimental results confirmed that accounting for the material's anisotropy can enhance the stiffness of structures made from short-reinforced composite materials. The present work aims to obtain a minimum compliance design through topology optimization, utilizing anisotropic material properties obtained from numerical simulation of the injection molding process. The Tsai–Hill failure criterion is employed in current work solely for design evaluation purposes, but it could also be incorporated into the optimization problem, which is the focus of future work.

In instances where predetermined anisotropy was taken into consideration, the orientation of the reinforcing fibers led to stiffness increases ranging from 5 to 10% for the available composite materials and can be increased more in the presence of materials with greater anisotropy. In the case of molding truss-type structures, the stiffness increase was somewhat lower, ranging from 0.8 to 7.3%. This discrepancy arises from the fact that, in truss-type structures, the majority of the fibers align with the load-bearing elements of the truss, while the truss contributes less to the overall rigidity of the structure.

Furthermore, the results obtained using this technique were compared with those of topology-optimal structures. Thus, the increase in stiffness is also expressed as a percentage. In specific applications, especially within the aerospace industry, such an increase in stiffness can prove to be significantly advantageous.

This work demonstrates that the modified $C_K{}^{TH}$, in terms of the Tsai–Hill failure criterion, can be effectively employed in the design of fiber-reinforced polymer-based composite structures. It is worth noting that the $C_K$ approach can be formulated using other failure criteria as well. For instance, the Tsai–Wu failure criterion may be more appropriate when distinguishing between tension and compression strengths is crucial.

Currently, various options exist for assessing weight efficiency using dimensionless criteria. Nevertheless, this field has yet to achieve a comprehensive form, and the approach proposed in this work contributes to the advancement of dimensionless methods for assessing the quality of structures made from short-reinforced composite materials.

**Author Contributions:** Conceptualization, E.K. (Evgenii Kurkin); methodology, E.K. (Evgenii Kurkin); validation, E.K. (Evgenii Kurkin) and O.U.E.B.; software, E.K. (Evgenii Kurkin) and E.K. (Evgenii Kishov); formal analysis, O.U.E.B. and E.K. (Evgenii Kurkin); investigation E.K. (Evgenii Kurkin), O.L., E.K. (Evgenii Kurkin), and O.U.E.B.; data curation, O.U.E.B. and E.K. (Evgenii Kurkin); writing—original draft preparation, O.U.E.B. and E.K. (Evgenii Kurkin); writing—review and editing, O.U.E.B., E.K. (Evgenii Kurkin), O.L., and E.K. (Evgenii Kurkin); visualization, O.U.E.B. and O.L.; supervision, E.K. (Evgenii Kurkin); project administration, E.K. (Evgenii Kurkin); funding acquisition, E.K. (Evgenii Kurkin). All authors have read and agreed to the published version of the manuscript.

**Funding:** The research on the design quality of short-reinforced composite-material structures was funded by the Russian Science Foundation, grant number 22-79-10309.

**Data Availability Statement:** The data presented in this study are available on request from the corresponding author.

**Acknowledgments:** We would like to express our special thanks of gratitude to Valery Komarov for advancing the use of dimensionless numbers in structural design and structural engineering theory. We would like to express our gratitude to Igor Slovtsov and Valery Komarov for bringing to our attention the potential usage of SFRC in aerospace structures. We would also like to extend our gratitude to Aleksandr Pavlov and Stanislav Selivanov for their support in performing the tensile test and to Pladep Ltd. for their support during injection molding.

**Conflicts of Interest:** The authors declare no conflicts of interest. The funders had no role in the design of the study, in the collection, analysis, or interpretation of data, in the writing of the manuscript, or in the decision to publish the results.

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
