# Peer review of "Topology Optimization and Efficiency Evaluation of Short-Fiber-Reinforced Composite Structures Considering Anisotropy"

_computation, doi:10.3390/computation12020035_

Round 1
Reviewer 1 Report
Comments and Suggestions for Authors
Please see attached file.

Author Response
We are deeply grateful to the reviewer for the insightful analysis and valuable recommendations for revision of the manuscript. The point-to-point responses to are attached below:
1) The reviewer considers that the abstract could collect in a more detailed way and closer to the reality of the He would avoid using the expressions “fluid-structure interaction” and “The materials used were non-linearly modelled” because the reader may confuse the scope of the work. If the reviewer has not misunderstood, a simultaneous simulation that encompasses fluid and structural mechanics is not carried out, but rather a first moulding simulation is carried out based on fluid mechanics, from which parameters required to model the elastic behaviour of the composite material, so there is no fluid-structure interaction, but a iteration of fluid and mechanical simulations performed in a serial way. Regarding the non-linear model of the material, the topological optimization carried out is linear, which should be clear, while the non- linear modelling is carried out to verify the already optimized design.
We improve the Abstract according the recommendations:
“Each iteration of topology optimization involves two consecutive steps: the first one is a simulation of the injection molding process for obtaining the fiber orientation tensor and the second is a structural analysis with anisotropic material properties.”
“Structural analysis using linear anisotropic material model was employed within the topology optimization. For verification, a non-linear elasto-plastic material model was used based on an exponential-and-linear hardening law.”
We hope that in this form the abstract becomes closer to the presented work.
2) 147, the expression “the reaction becomes nonlinear” may be improved: “the behaviour / stress-strain response becomes nonlinear”?
We have taken your recommendation into account and corrected this phrase: "the stress-strain response becomes nonlinear, and plastic deformation occurs".
3) 5, I would recommend describing the terms present in the equations in lines 173-181 in more detail, since they may be difficult to interpret for a reader who is not an expert in this specific topic. The “retain volume” could be defined as the minimum achievable volume, while the constraints g4 and g5 could be explained physically or referenced to previous works. Do I understand that the g4 allows setting a minimum width of structural elements on an average? While the restriction g5 is due to the fact that the densities vary in the thickness of the piece?
We've detailed the meaning of this constraints:
“Minimum member size” constraint g2(x) limits the change of gradient of density field with respect to spatial coordinates over the design domain. Thus, the minimum width of structural members an average becomes limited for specified value. “Pull-out direction” constraint g3(x) provides to monotone decreasing of density ρ(x) when moving away from a parting plane.
And we’ve moved the topological density with its range to the design variable section.
4) Regarding densities, the constraints are set between 0 and 1, but it is not mentioned if any projection/filtering process is carried out to avoid intermediate values? Or are intermediate values desirable? Can the manufacturing process be subsequently controlled to produce intermediate densities? Please clarify this issue.
Relationship between topological density and finite element stiffness matrix is carried out by the SIMP. For manufacturing purposes elements with density below specified threshold value were removed from the model and retained elements with density above threshold value are considered as part.
We mentioned these points in Methods 2.2.1. Topology optimization section.
5) Pag. 9. The fixture developed to carry out the experimental test is interesting, including plates to prevent out-of-plane buckling (preventing displacements in Z). However, is it verified that buckling does not occur in the XY plane in structural elements subjected to compression?
Out-of-plane buckling phenomena was discovered at the tooling design stage during linear buckling analysis. We have added the results of these calculations to the text of the manuscript:
2.2.3. Section: “The necessity of using the Z-displacement restricting plates is caused by performed linear buckling analysis.”
3.1.3. Section: “The linear buckling analysis shows that adding of the Z-displacement restricting plates allows to increase load multiplier from 0.8 to 1.3 for plastic parts and prevent them from buckling.”
3.2.3. Section: “The linear buckling analysis shows that adding of the Z-displacement restricting plates allows to increase load multiplier from 0.52 to 1.3 for plastic parts and prevent them from buckling.“
We didn't find any problems with in-plane buckling neither in the calculations nor in the subsequent experiments.
6) It is not clear how the boundary conditions are applied to the Is the applied load distributed uniformly in all the nodes that form the contact circumference? Has the effect of these conditions been verified?
At the stage of topological optimization linear boundary conditions are used, while at the verification calculation boundary conditions are set through auxiliary bodies, through one-sided contacts. We have described the boundary conditions in detail before each calculation result:
In 3.1.1: “During the topology optimization analysis a bearing force is used as loading boundary condition and a cylindrical supports are used as structural constraints.”
“Load and supports are applied to auxiliary cylindrical bodies which are located inside holes and connected to main part using nonlinear contact.“
In 3.2.1.: “For topology optimization stage linear static analysis is performed. Fixed supports are used as structural constraints and uniform load is applied directly to nodes of frozen elements.”
“For internal-exported and baseline designs loads and supports are the same as for topology optimization analysis. For reconstructed cases load and supports are applied to auxiliary cylindrical bodies which are modeled inside holes and connected to main part using nonlinear contact. Aluminum bushings are connected to the plastic parts by a linear bonded contact.”
7) Nothing is said about mesh sensitivity analysis, has any study been carried out in this regard?
We did not conduct mesh sensitivity analysis for topology optimization because a minimum member size constraint has been used in problem formulation. Corresponding sentence is added to section 2.2.1:
“In addition, its known [53] that minimum member size constraint acts as mesh independency filter for topology optimization.”
At the same time, we have previously investigated the mesh sensitivity of molding simulation, as we additionally mentioned in the article in section 3.1.2.:
“Mesh sensitivity analysis showed that the molding simulation gives almost the same results at coarse reduced meshes and fine meshes built on CAD models [82]”
8) In Figure 9, a significant change is observed between the experimental and numerical What is this due to? The effect of the adhesion of the aluminium bushings and other imperfections is then discussed, but it is not clear if it affects these results or only the 3D results.
The effect of adhesion is only exist in the Variable molding cases (previously named 3D) because bushes exists only there.
In the case of the mentioned figure, it seems to us that the plastic model overestimates the stiffness of the material in the area of the lugs, as well as the bolts gaps, so we have added such words in the manuscript:
“The difference between experimental and numerical results is due to overestimating a stiffness of bolted joints and rough approximation of testing tool flexibility.”
9) Lines 382-382: the expression “non-linear Structural analysis” could be Does it refer only to the plastic behaviour of the material or also to geometric non-linearity (large displacements/strains)?
Non-linear Structural analysis refer to plastic behaviour of the material and frictionless nonlinear contact in areas of bolted connection.
We have described in detail, mentioning it for each case, that the topological optimization stage uses linear calculation, whereas the verification stage uses nonlinear anisotropic material and nonlinear contacts.
In 3.1.1: “For topology optimization a linear static analysis is used.”
“At the verification analysis an anisotropic elasto-plastic material model is used considering orientation of reinforcing fibers, calculated for molded plate from which the parts are cut out.“
In 3.2.1.: “For topology optimization stage linear static analysis is performed. Fixed supports are used as structural constraints and uniform load is applied directly to nodes of frozen elements.”
“Verification was performed by nonlinear analysis with anisotropic elasto-plastic material model, considering injection molding simulation.” “For reconstructed cases load and supports are applied to auxiliary cylindrical bodies which are modeled inside holes and connected to main part using nonlinear contact. Aluminum bushings are connected to the plastic parts by a linear bonded contact.”
In our simulations Large displacements have no notable influence in the current cases.
10) Lines 538-539. The reviewer does not fully understand why the specific stiffness does not significantly increase in the 3D It is true that the effect of the orientation of fibres will be sensitive to the 3D approach, but the modelled structure is still a planar case in loads and boundary conditions (and the geometry is almost assimilable to a planar structure), a more detailed explanation of this would be appreciated for this interesting conclusion.
We improve the naming of optimization cases. We realized that 2D and 3D names not fully represents the actual situation. We replaced that to ”constant molding” and “variable molding” cases, and detailed what we put into these terms in 2.2.1. section:
“There are two manufacturing methods for composite parts production. The first method related to cutting a part from molded workpiece which can be for example in form of rectangular plate. This case named as “constant molding” where material anisotropy exists but not depend on the part shape. The second method is injection molding of designed part. The second case named as “variable molding” where a coupled simulation of injection molding and structural optimization is required.”
We also added a more detailed explanation of optimization results in Discussion section:
“In given work all resulting topologies correspond to truss structures, which consists mostly from rods. In variable molding case all rods has almost the same elastic properties due to alignment of fibers along it axis during injection molding.”
“In case of thin-walled structures, an effect of increasing stiffness considering variable molding anisotropy can be more prominent.”
Reviewer 2 Report
Comments and Suggestions for Authors
The manuscript implemented topology optimization for short-fiber 2 reinforced composite structures considering anisotropy. The conclusion is that considering anisotropy leads to stiffer structures and structural elements should be oriented in the direction of maximal stiffness. There are some queries needed to be clarified.
1. In the optimization model in Section 2.2.1, only stiffness is included. But in the manuscript, failure criterion is also considered and used as a standard to evaluate the results in Section 3. How is the failure criterion related to the optimization process? Why you use failure criterion to do the evaluation but not include it in the optimization?
2. TCI, TCA, TVI and TVA are used. But it is not clear how the differences are. For example, for TCI, the topology is also involved in the optimization. Why use the term topology constant? Also it is not clear how the performances are different, and how the optimization model affect the result.
3. Need to provide a flow chart to clarify how the whole process is implemented.
4. It seems that including anisotropy doesn’t have a big improvement on the structural stiffness. The authors give the value of less than 10% in the conclusion. The authors mentioned that the reason may be caused by the truss structure. But actually, the isotropic truss and the fiber oriented truss will have big differences on the stiffness since the match of the material orientation with the structure can high increase the structural stiffness. It is not reasonable to give this conclusion. It is suggested to increase the difference between E1 and E2 to see whether the small increase is caused by the small differences of E1 and E2 for short fiber reinforced composites. Also find other reasons.
Comments on the Quality of English Language
Suggest minor modifications on the quality of English language.
Author Response
We thank the reviewer for the attention given to our manuscript and valuable recommendations for its improvement. The point-to-point responses are presented below:
- In the optimization model in Section 2.2.1, only stiffness is included. But in the manuscript, failure criterion is also considered and used as a standard to evaluate the results in Section 3. How is the failure criterion related to the optimization process? Why you use failure criterion to do the evaluation but not include it in the optimization?
There are several possible formulations of the topological optimization problem, including the problem of minimizing the compliance at a given achievable volume constraint and the problem of minimizing the mass of a structure under constraints on its load carrying capacity. Within the current manuscript, we have considered the solution to the first of these statements, since it is the basic one. In the current work, we have evaluated the contribution of the anisotropic stiffness matrix to the compliance minimization problem.
We wish to consider in future research the case of a mass minimization problem formulation under constraints on the load carrying capacity of a structure, for which it will be necessary to consider the Tsai-Hill criterion in the topological optimization loop.
We thank the reviewer for such a valuable recommendation. We ask that the current problem statement be viewed as a separate task and a step toward a more complex problem. To be accurate and to clearly indicate the boundaries of our study we have made refinements by adding the following paragraphs to the methods and conclusion of the manuscript:
In Methods (2.2.2): “It should be mentioned that failure criteria presented above are not considered in topology optimization problem statement since the primary scope of given work is to investigate an influence of anisotropic properties of SFRP on classical topology optimization results aimed to obtain minimum compliance design.”
In Conclusion: “Present work aimed to obtain minimum compliance design using topology optimization with anisotropic material properties obtained from numerical simulation of injection molding process. Tsai-Hill failure criterion used in current work only for design evaluation, but it can be also included in optimization problem statement which is the subject of the future work.”
- TCI, TCA, TVI and TVA are used. But it is not clear how the differences are. For example, for TCI, the topology is also involved in the optimization. Why use the term topology constant? Also it is not clear how the performances are different, and how the optimization model affect the result.
We clarified meaning of the TCI, TCA, TVI and TVA abbreviations:
“TCA (Topology optimized design with Constant molding and Anisotropic material) and TCI (Topology optimized design with Constant molding and Isotropic material) cases, respectively. Here “constant molding” means that a single injection molding simulation was performed before optimization and the resulting solid material properties remains the same over the whole design domain.”
“TVA (Topology optimized design with Variable molding and Anisotropic material) and TVI (Topology optimized design with Variable molding and Isotropic material), respectively. Here “variable molding” means that the injection molding simulation performed at each optimization cycle, and solid material properties are updated according to orientation tensor field redistribution.”
- Need to provide a flow chart to clarify how the whole process is implemented.
Thank you for the recommendation, we have added a flow chart on figure 2.
- It seems that including anisotropy doesn’t have a big improvement on the structural stiffness. The authors give the value of less than 10% in the conclusion. The authors mentioned that the reason may be caused by the truss structure. But actually, the isotropic truss and the fiber oriented truss will have big differences on the stiffness since the match of the material orientation with the structure can high increase the structural stiffness. It is not reasonable to give this conclusion. It is suggested to increase the difference between E1 and E2 to see whether the small increase is caused by the small differences of E1 and E2 for short fiber reinforced composites. Also find other reasons.
Thank you for the recommendation. We have added a separate section 3.1.2 comparing the influence of the ratio E1 to E2 on the shape and specific stiffness of structures.
In variable anisotropy cases fibers are oriented along structural members axis, which reduces the influence of anisotropy. We add one more reason to Discussion about this point:
“In case of thin-walled structures, an effect of increasing stiffness considering variable molding anisotropy can be more prominent.”
We have also added the summary of the new section to the Discussion
"Increasing the E1 to E2 ratio to 7.4 enables a 49% increase in the stiffness of the structure in constant molding case, which shows the possibility of increasing the effect in the presence of materials with greater anisotropy. "
and Conclusion
"... and can be increased more in the presence of materials with greater anisotropy."
Round 2
Reviewer 1 Report
Comments and Suggestions for Authors
Reviewer would like to thanks the authors for all the clarifications and explanations. Apart from correcting all the remarks from previous review, authors have given new conclusions and completed some details that give more value to the manuscript. Congratulations.
Reviewer 2 Report
Comments and Suggestions for Authors
It is suggested to accept the manuscript in the current version.